# Post-Anthesis Nitrogen Dynamic Models and Characteristics of Rice Combined with Sowing Date and Nitrogen Application Rate

Ying Ye [1,2], Kaocheng Zhao [1,2], Jun Ma [1,2], Lifen Huang [1,2] and Hengyang Zhuang [1,2,*]

1   Jiangsu Key Laboratory of Crop Genetics and Physiology/Jiangsu Key Laboratory of Crop Cultivation and Physiology, Agricultural College of Yangzhou University, Yangzhou 225009, China; mz120190931@yzu.edu.cn (Y.Y.); mz120190954@yzu.edu.cn (K.Z.); mx120190608@yzu.edu.cn (J.M.); lfhuang@yzu.edu.cn (L.H.)
2   Jiangsu Co-Innovation Center for Modern Production Technology of Grain Crops, Yangzhou University, Yangzhou 225009, China
*   Correspondence: hyzhuang@yzu.edu.cn

**Abstract:** In order to explore the effect of the relationship between the combination of sowing date and nitrogen application rate on the nitrogen status of rice plants and nitrogen uptake and transfer after anthesis, three sowing dates were set—23 May (S1), 2 June (S2), and 12 June (S3)—and four nitrogen fertilizer treatments—no nitrogen (N0), 180 (N1), 270 (N2), and 360 (N3) kg N/hm$^2$—were applied in a field experiment. The dynamic characteristics of nitrogen in rice post-anthesis under different treatments were analyzed by model fitting. The results showed that the three-leaf SPAD values of rice under different treatments varied, exhibiting a slow–fast–slow inverted S-shaped curve on the days after anthesis. However, the maximum SPAD value ($k_s$), the time to enter the rapid period of decline ($t1_s$), and the time to reach the maximum rate ($T_s$) were different between the different treatments. The maximum SPAD ($k_s$) values of each sowing date increased with the increase in nitrogen fertilizer application; the $t1_s$ of each treatment was 15–29 days after spike development, and the S3 treatment entered the rapid decline period the earliest. It was beneficial to the transfer of leaf nitrogen to grain, and the nitrogen content, dry matter, and nitrogen uptake of stem sheaths under different treatments varied with days after anthesis; the S3 treatment exhibited the highest values. Leaf nitrogen content and dry matter decreased linearly in different treatments, and leaf nitrogen uptake showed an exponential downward trend. The parameters $a_{lnc}$, $a_{ldm}$, $a_{lnu}$, $b_{lnc}$, $b_{ldm}$, and $b_{lnu}$ all increased gradually with the delay in sowing date and the increase in nitrogen, and the maximum values were obtained in the S3N3 treatment. The dry matter accumulation and nitrogen uptake of all treated grains showed a slow–fast–slow S-shaped upward trend over time; the maximum dry matter accumulation ($k_{gdm}$) of grains was the greatest at 9652.7 kg/hm$^2$, and the duration of the rapid grouting period ($t2_{gdm} - t1_{gdm}$) was the longest, lasting 32 days. The maximum nitrogen absorption of grains ($k_{gnu}$) was highest in the S3N3 treatment, whereas the rapid nitrogen absorption duration of grains ($t2_{gnu} - t1_{gnu}$) was the longest in S1N0. These results provide a basis for the development of optimized nitrogen fertilizer application, real-time nitrogen fertilizer management, and post-anthesis nitrogen uptake and distribution models for rice with different sowing dates.

**Keywords:** planting times; dry matter; nitrogen accumulation and translocation; post-anthesis dynamic models





## 1. Introduction

Rice is one of the most important food crops in the world, meeting the dietary needs of more than half of the global population and playing a decisive role in ensuring food security [1,2]. A large number of experimental results show that rice yield is closely related to sowing date [3,4] and nitrogen application [5,6]. The sowing date mainly affects the growth and development of rice through climate change [7–10]. Warm climate has

accelerated the growth and development of rice and has shortened the growth period [11]. The average temperature during grain filling regulates the grouting rate [12]. Reasonable determination of the sowing date to keep the rice fruiting period in a better light and temperature state is a key technique in cultivation management, and it is also the basis for the high yield and quality of rice [13]. Nitrogen is one of the determinants of rice growth, development, and yield. Studies have shown that nitrogen uptake of rice interacts with its biomass accumulation, leaf area, and anthesis, and the application of nitrogen fertilizer can increase the nitrogen accumulation of aboveground dry matter, thereby achieving the effect of increasing yield; however, excessive application of nitrogen fertilizer and low nitrogen use efficiency will reduce rice yield [14–19].

The nitrogen status and dynamic changes in rice during flowering affect not only yield but also nitrogen status, stress resistance, and physiological efficiency [20]. The post-anthesis of rice is a critical period for panicle growth and development, and improving dry matter accumulation and nitrogen uptake in rice after anthesis is the key to nitrogen use efficiency [21]. The production of photosynthate after anthesis shows the relationship between population status and environment, including the complexity and distribution process of nitrogen uptake after anthesis determined by light and temperature [22,23]. To obtain the optimal nitrogen application scheme that results in high yield while balancing the above-mentioned factors, the dynamic changes in nitrogen after rice anthesis should be continuously monitored [24]. Due to the complexity and uncertainty of field tests, it is difficult to conduct field tests over a long duration. The crop growth model can present the basic laws and quantitative relationships of the crop growth and development process and quantitatively predict the dynamic behavior of the crop growth system [25].

At present, the nitrogen model of rice is mainly based on critical nitrogen concentration [26,27] and nutritional diagnosis [28,29]. There is little research on the nitrogen dynamic model after anthesis under the combination of sowing date and nitrogen application rate. Therefore, in this experiment, carried out in Jiangsu Province, the high-quality rice variety Nanjing 9108 was used as the material [30]. The experiment analyzes the changes in nitrogen dynamic characteristic parameters and their relationship with the utilization efficiency of rice after anthesis, and monitors and manages nitrogen absorption and utilization in real time.

## 2. Materials and Methods

### 2.1. Experimental Site

The test was conducted from May to November 2018 at the Experimental Farm of the College of Agronomy of Yangzhou University, Yangzhou city, Jiangsu Province, China (119°25′ E, 32°23′ N), with an average annual precipitation of 1288 mm and an average sunshine duration of 1973.9 h. The stubble in the test field was wheat, and the soil type was sandy loam, with cultivated organic matter at 18.76 g/kg, total nitrogen at 1.26 g/kg, alkalized nitrogen at 0.08 g/kg, available phosphorus at 0.03 g/kg, and available potassium at 0.09 g/kg.

### 2.2. Experimental Design

The test material was Nanjing 9108 (late-ripening japonica). A randomized block design was established; three sowing dates were expressed as S1, S2, and S3, and nitrogen was applied at 0, 180, 270, and 360 kg N/hm$^2$ on 23 May, 2 June, and 12 June, with two repetitions. The experimental area was 6.6 m long, 3.9 m wide, and 25.74 m$^2$ in size. Prior to sowing, calcium superphosphate 450 kg/hm$^2$ and potassium chloride 150 kg/hm$^2$ were applied to each treatment. The nitrogen fertilizer tested was urea, and the fertilizer ratio of basis: first tiller: second tiller: flower promotion: flower preservation was 6:3:3:5:3. The time of the application of the basal fertilizer on each sowing date was the day before the seedlings were transplanted. We transplanted seedlings with 3.4 and 3.8 leaves, with a plant spacing of 30 cm × 13 cm and four seedlings per hole. Other field management in the test process was the same as field production.

*2.3. Determination Methods*

2.3.1. Determination of the Leaf SPAD Value

A representative 10 main stems of rice were measured for each treatment. Every 7 days after anthesis, the SPAD value of three leaves were measured with a SPAD-502 chlorophyll meter manufactured by Konica Minolta, measuring the middle of each leaf (avoiding the main veins) until maturity [31].

2.3.2. Determination of the Plant Dry Matter

After rice anthesis, four representative holes were taken every seven days for each treatment, and the plants were divided into three parts: stem sheath, leaf, and grain. The samples were placed in an oven at 105 °C for 30 min and then dried at 80 °C to constant weight. After cooling to room temperature in a dry environment, the dry matter was measured.

2.3.3. Total Nitrogen Determination

The dried stem sheath, leaves, and grains were crushed separately. After passing them through a 100-mesh sieve, 0.25 g of the samples was collected, and the nitrogen concentration of each plant organ was determined by the $H_2SO_4$–$H_2O_2$ combined cooking and the semi-trace Käger distillation method [32].

*2.4. Index Calculation*

The nitrogen absorption of each organ of the plant = the nitrogen content of each organ of the plant × the accumulation of dry matter of each organ of the plant.

*2.5. Model and Characteristic Parameters*

2.5.1. Logistic Model

The mathematical formula of the logistic model is as follows:

$$y = k/(1 + ae^{-bx}) \tag{1}$$

The growth rate equation of the first derivative of Equation (1) was found with Equation (2):

$$dy/dx = kabe^{-bx}/(1 + ae^{-bx})^2 \tag{2}$$

The inflection points (t1) and (t2) of the growth rate, the maximum rate, and up to the maximum rate of descent time (T, d) were obtained using Equation (2).

$$t1 = -(\ln(2 + \sqrt{3})/a)/b \tag{3}$$

$$t2 = -(\ln(2 - \sqrt{3})/a)/b \tag{4}$$

$$T = \ln a/b \tag{5}$$

2.5.2. A Reverse Logistic Model

The mathematical formula of the reverse logistic model is as follows:

$$y = c - k/(1 + ae^{-bx}) \tag{6}$$

The first derivative was found based on the above formula, and Equation (2) was obtained. By calculating the first and second derivatives using Equation (2), the inflection points (t1, d) and (t2, d) and up to the maximum rate of descent time (T, d) were determined.

*2.6. Data Processing Methods*

Microsoft Excel 2010 was used for data processing, IBM SPSS Statistics 23.0 for data analysis, and Origin 9.0 for curve simulation.

## 3. Results

### 3.1. Dynamics of the SPAD Value of the Upper Three Leaves of Rice Combined with the Sowing Date and Nitrogen Application Rate

The curve fitting of the SPAD value of the upper three leaves with days after anthesis under different treatments is shown in Figures 1–3; the SPAD value changed over time, and all treatments showed the characteristics of an inverted S-shaped decline, which could be fit to Formula (6). The main characteristic parameters of the dynamic change in the SPAD value of the upper three leaves in each treatment were calculated by Equations (2)–(5).

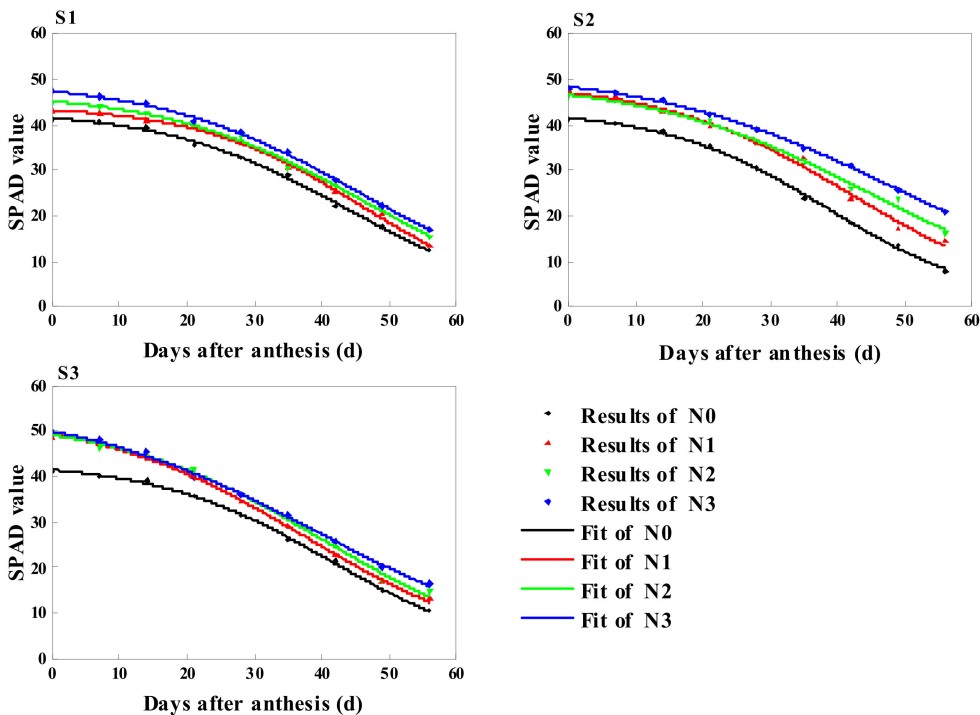

**Figure 1.** Change in SPAD values of inverted sword leaves (days after anthesis).

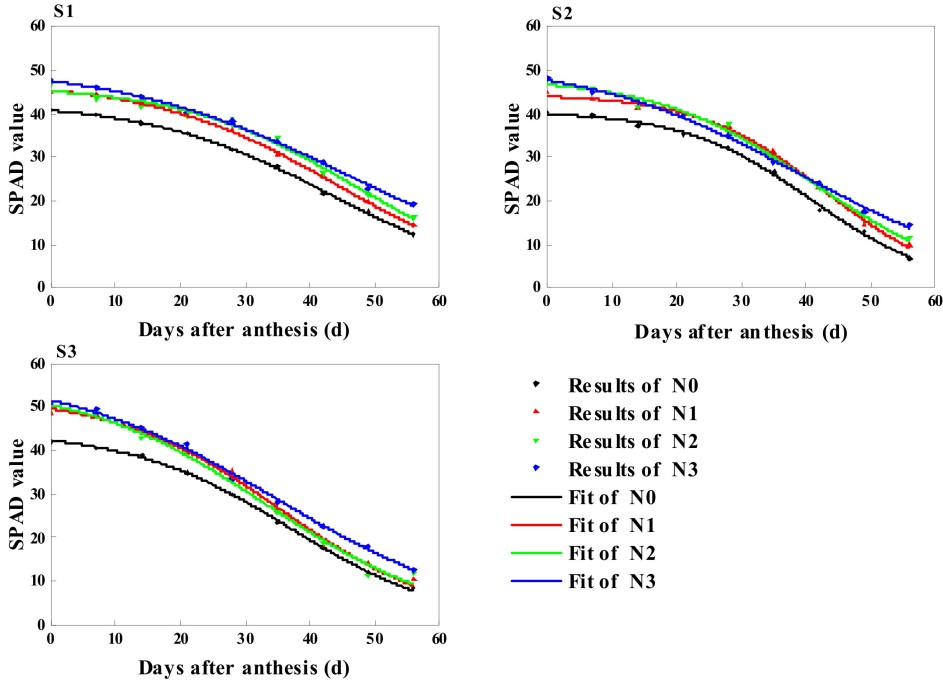

**Figure 2.** Change in SPAD values of inverted second leaves (days after anthesis).

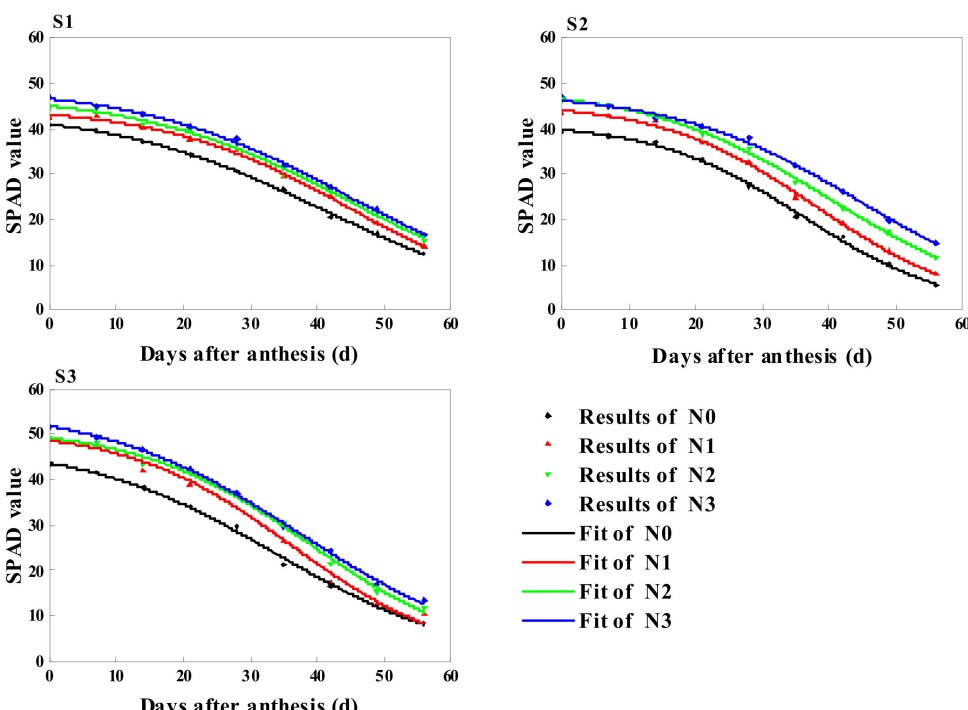

**Figure 3.** Change in SPAD values of inverted third leaves (days after anthesis).

There were significant differences in the main parameters of dynamic changes in the SPAD value of the upper three leaves after anthesis, as shown in Table 1. The coefficient of determination $R^2$ of each treatment curve was above 0.987, indicating that the curve fitting results were ideal. The maximum SPAD value ($k_s$), the inflection point time ($t1_s$) associated with the growth rate, and the time to reach the maximum rate ($T_s$, d) were different under different treatments. With the increase in nitrogen fertilizer, the $k_s$ of different sowing dates gradually increased; the time for each treatment to enter the rapid descending period was 15–29 d, and the earliest rapid descent period of the upper three leaves occurred in the S3 treatment. However, the longest duration of the rapid descent period of inverted sword leaf was observed in the S2 treatment, and the inverted second leaf and inverted third leaf were ranked as S1 > S2 > S3. In addition to the N0 level, the duration of the rapid descent period in each treatment showed an upward trend with the increase in nitrogen fertilizer, indicating that the higher the nitrogen application, the longer and slower the rapid decline of foliar nitrogen. The value of $t1_s$ in the S3 treatment was minimal, which was conducive to leaf nitrogen transfer to grains, and the time to obtain the maximum rate in different treatments was 31–47 d. Under the interaction of the sowing date and nitrogen application, $t1_s$ and $T_s$ were ranked as inverted sword leaf > inverted second leaf > inverted third leaf, which was due to the earlier transfer of nitrogen to reproductive organs in the lower leaves of rice.

**Table 1.** Characteristic parameters of fitted equation for the SPAD value of the upper three leaves ($y = c - k/(1 + ae^{-bx})$). These values represent the arithmetic mean (average) of two duplicate values, with $\pm$ representing the standard error. Lowercase letters indicate significant differences ($p$-value < 0.05) in the SPAD values at different nitrogen application rates within each sowing date.

| Treatment | | Leaf | $R^2$ | $k_s$ | $a_s$ | $b_s$ | $c_s$ | $t1_s$ | $T_s$ |
|---|---|---|---|---|---|---|---|---|---|
| Sowing Dates | Nitrogen Levels | | | | | | | | |
| S1 | N0 | | 0.998 | 41.08 ± 0.34 c | 26.17 ± 0.07 b | 0.077 ± 0.001 a | 42.86 ± 0.75 b | 25.46 ± 0.13 b | 42.68 ± 0.25 a |
| | N1 | | 0.997 | 43.41 ± 0.16 b | 43.67 ± 0.99 a | 0.083 ± 0.001 a | 44.12 ± 0.25 b | 29.82 ± 0.10 a | 45.78 ± 0.01 a |
| | N2 | | 0.996 | 48.56 ± 0.28 a | 22.62 ± 0.40 bc | 0.067 ± 0.003 b | 47.20 ± 0.54 a | 27.13 ± 0.76 b | 46.96 ± 1.50 a |
| | N3 | | 0.998 | 49.80 ± 0.10 a | 20.55 ± 0.66 c | 0.066 ± 0.002 b | 49.63 ± 0.12 a | 26.04 ± 0.11 b | 46.15 ± 0.57 a |
| S2 | N0 | Inverted sword leaf | 0.998 | 42.19 ± 0.26 d | 25.18 ± 0.72 a | 0.085 ± 0.004 a | 43.04 ± 0.67 b | 22.61 ± 0.60 a | 38.23 ± 1.25 c |
| | N1 | | 0.993 | 43.42 ± 0.10 c | 26.14 ± 1.54 a | 0.083 ± 0.000 a | 48.33 ± 0.17 a | 23.43 ± 0.71 a | 39.29 ± 0.71 bc |
| | N2 | | 0.991 | 47.51 ± 0.24 b | 15.29 ± 0.55 b | 0.063 ± 0.001 b | 49.44 ± 1.45 a | 22.56 ± 0.76 a | 43.63 ± 0.92 ab |
| | N3 | | 0.998 | 49.58 ± 0.04 a | 14.73 ± 1.96 b | 0.057 ± 0.003 b | 51.48 ± 0.61 a | 24.07 ± 1.30 a | 47.43 ± 0.27 a |
| S3 | N0 | | 0.998 | 41.88 ± 0.46 c | 26.14 ± 2.52 a | 0.080 ± 0.001 a | 42.94 ± 1.02 b | 24.29 ± 1.51 a | 40.76 ± 1.72 a |
| | N1 | | 0.999 | 45.26 ± 0.42 b | 16.50 ± 0.81 ab | 0.081 ± 0.001 a | 51.45 ± 0.21 a | 18.45 ± 0.50 ab | 34.81 ± 0.39 a |
| | N2 | | 0.996 | 46.70 ± 0.09 ab | 17.58 ± 2.17 ab | 0.076 ± 0.001 a | 51.37 ± 0.04 a | 20.27 ± 1.37 ab | 37.60 ± 1.14 a |
| | N3 | | 0.997 | 48.41 ± 0.01 a | 10.51 ± 0.86 b | 0.064 ± 0.001 b | 53.93 ± 0.36 a | 16.10 ± 1.02 b | 36.69 ± 0.70 a |
| S1 | N0 | | 0.999 | 42.79 ± 0.08 d | 21.77 ± 2.65 bc | 0.071 ± 0.001 a | 42.62 ± 0.55 c | 24.72 ± 1.38 b | 43.27 ± 1.12 b |
| | N1 | | 0.996 | 44.80 ± 0.10 c | 24.42 ± 1.23 ab | 0.075 ± 0.002 a | 46.98 ± 0.31 b | 25.20 ± 0.17 b | 42.88 ± 0.19 b |
| | N2 | | 0.987 | 46.92 ± 0.45 b | 31.85 ± 1.51 a | 0.073 ± 0.001 a | 46.55 ± 0.25 b | 29.35 ± 0.25 a | 47.40 ± 0.00 a |
| | N3 | | 0.997 | 48.72 ± 0.08 a | 13.86 ± 0.70 c | 0.058 ± 0.002 b | 50.66 ± 0.16 a | 22.60 ± 0.09 b | 45.32 ± 0.69 ab |
| S2 | N0 | Inverted second leaf | 0.997 | 41.09 ± 0.06 d | 59.66 ± 4.88 b | 0.100 ± 0.002 a | 40.60 ± 0.18 d | 27.68 ± 0.27 ab | 40.85 ± 0.00 a |
| | N1 | | 0.997 | 43.53 ± 0.29 c | 87.98 ± 3.55 a | 0.106 ± 0.003 a | 44.31 ± 0.43 c | 29.97 ± 1.09 a | 42.46 ± 1.39 a |
| | N2 | | 0.994 | 46.65 ± 0.03 b | 32.47 ± 0.81 c | 0.086 ± 0.002 b | 47.92 ± 0.31 b | 25.16 ± 0.30 b | 40.48 ± 0.65 a |
| | N3 | | 0.995 | 48.86 ± 0.07 a | 11.88 ± 0.59 d | 0.066 ± 0.003 c | 51.34 ± 1.05 a | 17.65 ± 0.08 c | 37.79 ± 0.69 a |
| S3 | N0 | | 0.999 | 40.64 ± 0.31 d | 26.59 ± 0.48 b | 0.092 ± 0.003 b | 43.45 ± 0.18 c | 21.37 ± 0.89 a | 35.70 ± 1.36 a |
| | N1 | | 0.998 | 42.81 ± 0.02 c | 34.33 ± 1.03 a | 0.108 ± 0.002 a | 49.92 ± 0.27 b | 20.64 ± 0.01 ab | 32.89 ± 0.18 ab |
| | N2 | | 0.990 | 45.01 ± 0.39 b | 19.87 ± 1.53 c | 0.096 ± 0.003 ab | 51.76 ± 0.08 b | 17.38 ± 0.26 bc | 31.11 ± 0.17 b |
| | N3 | | 0.997 | 48.70 ± 0.18 a | 11.62 ± 0.83 d | 0.076 ± 0.001 c | 54.96 ± 0.58 a | 15.00 ± 0.84 c | 32.45 ± 0.73 ab |
| S1 | N0 | | 0.997 | 41.17 ± 0.14 d | 14.79 ± 0.02 c | 0.068 ± 0.001 b | 43.46 ± 0.36 b | 20.26 ± 0.32 c | 39.63 ± 0.60 b |
| | N1 | | 0.997 | 42.05 ± 0.04 c | 27.78 ± 1.18 a | 0.076 ± 0.001 a | 44.42 ± 0.50 b | 26.40 ± 0.22 a | 43.73 ± 0.02 a |
| | N2 | | 0.992 | 45.57 ± 0.20 b | 18.92 ± 0.23 b | 0.066 ± 0.000 bc | 47.22 ± 0.39 a | 24.60 ± 0.18 ab | 44.55 ± 0.19 a |
| | N3 | | 0.997 | 49.12 ± 0.18 a | 17.28 ± 0.62 bc | 0.063 ± 0.000 c | 49.27 ± 0.50 a | 24.32 ± 0.57 b | 45.22 ± 0.57 a |
| S2 | N0 | Inverted third leaf | 0.997 | 41.51 ± 0.29 d | 25.50 ± 1.88 ab | 0.090 ± 0.002 a | 41.32 ± 0.29 c | 21.43 ± 0.46 b | 36.15 ± 0.22 b |
| | N1 | | 0.998 | 44.03 ± 0.03 c | 29.76 ± 0.29 a | 0.091 ± 0.003 a | 45.35 ± 0.18 b | 22.96 ± 0.53 b | 37.52 ± 0.93 b |
| | N2 | | 0.998 | 46.52 ± 0.16 b | 20.91 ± 0.88 b | 0.078 ± 0.002 b | 48.55 ± 0.49 a | 22.22 ± 0.11 b | 39.22 ± 0.22 b |
| | N3 | | 0.997 | 48.03 ± 0.14 a | 24.87 ± 0.16 ab | 0.072 ± 0.001 b | 47.97 ± 0.06 a | 26.34 ± 0.28 a | 44.64 ± 0.53 a |
| S3 | N0 | | 0.998 | 41.12 ± 0.06 c | 16.64 ± 0.35 c | 0.088 ± 0.001 b | 45.53 ± 0.31 d | 17.08 ± 0.34 b | 32.14 ± 0.43 c |
| | N1 | | 0.988 | 42.70 ± 0.27 c | 41.97 ± 1.59 a | 0.110 ± 0.001 a | 48.74 ± 0.06 c | 21.99 ± 0.14 a | 33.97 ± 0.04 bc |
| | N2 | | 0.996 | 45.90 ± 0.46 b | 29.22 ± 0.71 b | 0.091 ± 0.000 b | 50.28 ± 0.18 b | 22.60 ± 0.65 a | 37.07 ± 0.65 a |
| | N3 | | 0.999 | 49.30 ± 0.46 a | 16.48 ± 0.25 c | 0.079 ± 0.001 c | 54.25 ± 0.05 a | 18.92 ± 0.32 b | 35.70 ± 0.42 ab |

*3.2. Effects of the Combination of Sowing Date and Nitrogen Application on the Nitrogen Content of Rice*

3.2.1. Dynamics of the Stem-Sheath Nitrogen Content under Different Treatments

The change in the curve of the stem-sheath nitrogen content with days after anthesis is shown in Figure 4. The changes in the stem-sheath nitrogen content over time were the same as those in the stem-sheath dry matter. The quadratic function $y = ax + bx^2 + c$ was used for fitting, and a, b, and c were the parameters to be determined. The nitrogen contents of the stem sheath of three sowing dates were the highest at the N3 level, and the difference between the nitrogen levels was significant.

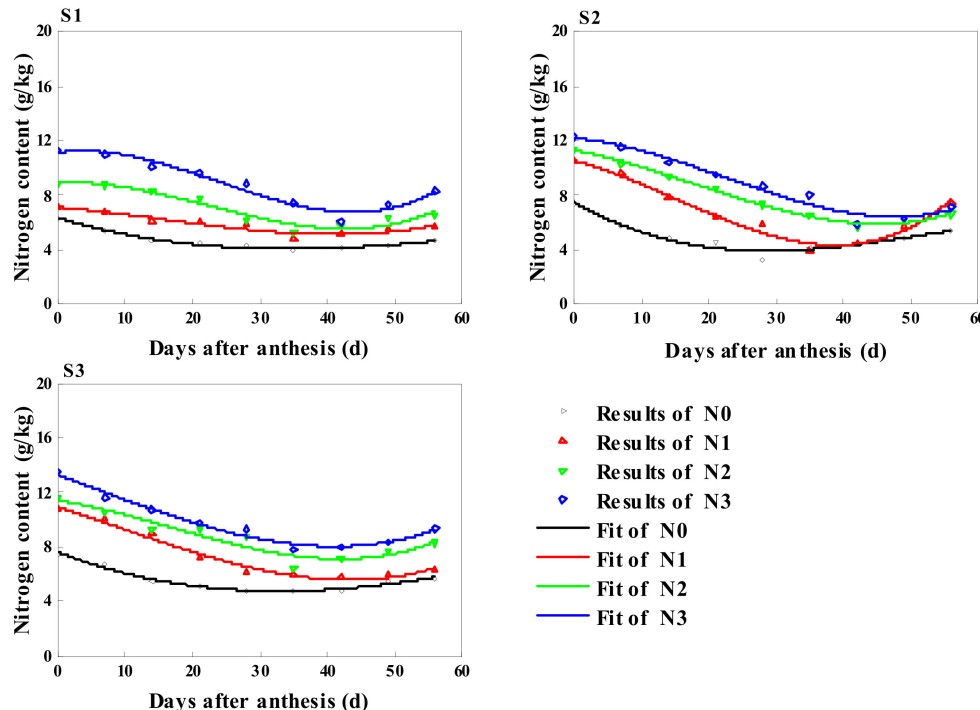

**Figure 4.** Change in nitrogen content of the stem-sheath (days after anthesis).

The main parameters of the dynamic change in the stem-sheath nitrogen content after anthesis were significantly different, except for $b_{snc}$, as shown in Table 2. The coefficient of determination of each treatment curve $R^2$ was between 0.815 and 0.972. The theoretical minimum points of the different treatments corresponded to days after anthesis ($X_{snc}$, d) and the minimum values ($Y_{snc}$, g/kg). The $X_{snc}$ values of S1 and S3 were the highest at the N3 level and $Y_{snc}$ gradually increased with the increase in nitrogen fertilizer, except at the N1 level; S2 at each nitrogen level was lower than S1 and S3; and the maximum value of 7.95 g/kg was obtained at the N3 level under S3.

**Table 2.** Characteristic parameters of the fitted equation for the stem-sheath nitrogen content ($y = ax + bx^2 + c$). These values represent the arithmetic mean (average) of two duplicate values, with $\pm$ representing the standard error. Lowercase letters indicate significant differences (*p*-value < 0.05) in the stem-sheath nitrogen content at different nitrogen application rates within each sowing date.

| Treatment | | | | | | | |
|---|---|---|---|---|---|---|---|
| Sowing Dates | Nitrogen Levels | $R^2$ | $a_{snc}$ | $b_{snc}$ | $c_{snc}$ | $X_{snc}$ | $Y_{snc}$ |
| S1 | N0 | 0.958 | $-0.124 \pm 0.008$ ab | $0.002 \pm 0.000$ a | $6.28 \pm 0.32$ c | $31.00 \pm 2.00$ b | $4.35 \pm 0.07$ c |
| | N1 | 0.838 | $-0.099 \pm 0.006$ a | $0.001 \pm 0.000$ a | $7.33 \pm 0.09$ c | $49.50 \pm 3.00$ a | $4.86 \pm 0.21$ c |
| | N2 | 0.815 | $-0.152 \pm 0.003$ bc | $0.002 \pm 0.000$ a | $9.50 \pm 0.03$ b | $38.00 \pm 0.75$ ab | $6.60 \pm 0.09$ b |
| | N3 | 0.825 | $-0.179 \pm 0.012$ c | $0.002 \pm 0.000$ a | $11.94 \pm 0.20$ a | $44.63 \pm 2.88$ a | $7.93 \pm 0.32$ a |
| S2 | N0 | 0.889 | $-0.208 \pm 0.004$ a | $0.003 \pm 0.000$ a | $7.26 \pm 0.15$ b | $34.59 \pm 0.59$ c | $3.67 \pm 0.03$ c |
| | N1 | 0.884 | $-0.353 \pm 0.003$ b | $0.005 \pm 0.000$ a | $11.27 \pm 0.50$ a | $35.25 \pm 0.25$ c | $5.06 \pm 0.41$ bc |
| | N2 | 0.966 | $-0.219 \pm 0.001$ a | $0.002 \pm 0.000$ a | $11.72 \pm 0.03$ a | $54.63 \pm 0.13$ a | $5.75 \pm 0.00$ b |
| | N3 | 0.933 | $-0.206 \pm 0.002$ a | $0.002 \pm 0.000$ a | $12.57 \pm 0.19$ a | $51.50 \pm 0.50$ b | $7.26 \pm 0.29$ a |
| S3 | N0 | 0.972 | $-0.164 \pm 0.005$ a | $0.002 \pm 0.000$ a | $7.54 \pm 0.06$ c | $40.88 \pm 1.13$ ab | $4.20 \pm 0.25$ c |
| | N1 | 0.942 | $-0.241 \pm 0.008$ bc | $0.003 \pm 0.000$ a | $11.18 \pm 0.08$ b | $40.08 \pm 1.25$ ab | $6.36 \pm 0.38$ b |
| | N2 | 0.837 | $-0.208 \pm 0.004$ b | $0.003 \pm 0.000$ a | $11.78 \pm 0.17$ b | $34.67 \pm 0.67$ b | $8.17 \pm 0.03$ a |
| | N3 | 0.962 | $-0.259 \pm 0.008$ c | $0.003 \pm 0.000$ a | $13.53 \pm 0.11$ a | $43.08 \pm 1.25$ a | $7.95 \pm 0.22$ a |

### 3.2.2. Dynamics of the Leaf Nitrogen Content under Different Treatments

The change in the leaf nitrogen content with days after anthesis is shown in Figure 5. The leaf nitrogen content was the same as the time change trend, i.e., a straight line; the linear function $y = ax + b$ was used for fitting, the absolute value of $a_{lnc}$ was the slope, and $b_{lnc}$ was the theoretical leaf nitrogen content at the panicle stage. The leaf nitrogen content showed an upward trend with the delay in sowing date and gradually increased with the increase in nitrogen level, and the difference between treatments decreased with the growth time.

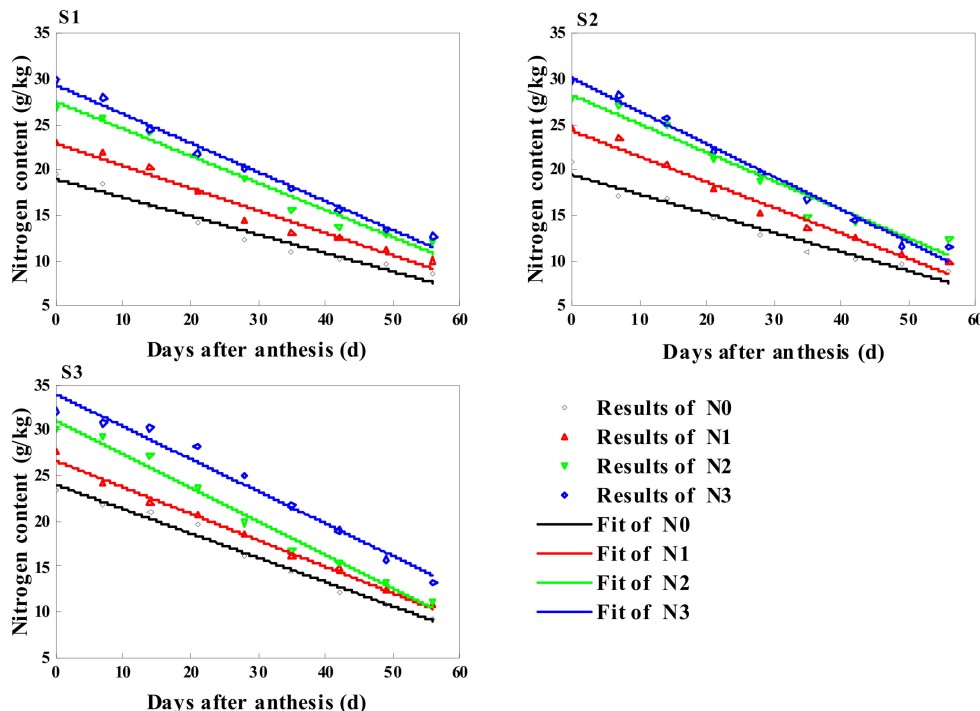

**Figure 5.** Change in nitrogen content of the leaves (days after anthesis).

The main parameters of the dynamic change in the leaf nitrogen content after anthesis varied significantly between the treatments, as shown in Table 3, and the coefficient of determination of each treatment curve $R^2$ was between 0.917 and 0.957. The absolute value of $a_{lnc}$ and $b_{lnc}$ gradually increased with the increase in nitrogen application, and the postponement of the $b_{lnc}$ follow-up period showed an upward trend. Under the interaction of sowing date and nitrogen application, $b_{lnc}$ was the highest in the S3 treatment, and the maximum value of 34.01 g/kg was obtained at the N3 level.

**Table 3.** Characteristic parameters of the fitted equation for the leaf nitrogen content (y = ax + b). These values represent the arithmetic mean (average) of two duplicate values, with $\pm$ representing the standard error. Lowercase letters indicate significant differences (*p*-value < 0.05) in the leaf nitrogen content at different nitrogen application rates within each sowing date.

| Treatment | | $R^2$ | $a_{lnc}$ | $b_{lnc}$ |
|---|---|---|---|---|
| **Sowing Date** | **Nitrogen Levels** | | | |
| | N0 | 0.935 | $-0.20 \pm 0.01$ a | $18.98 \pm 0.65$ c |
| S1 | N1 | 0.925 | $-0.25 \pm 0.01$ ab | $22.75 \pm 0.26$ b |
| | N2 | 0.941 | $-0.31 \pm 0.00$ b | $27.47 \pm 0.09$ a |
| | N3 | 0.957 | $-0.32 \pm 0.02$ b | $29.29 \pm 0.23$ a |
| | N0 | 0.944 | $-0.20 \pm 0.01$ a | $19.03 \pm 0.70$ c |
| S2 | N1 | 0.917 | $-0.28 \pm 0.02$ ab | $23.75 \pm 0.74$ b |
| | N2 | 0.948 | $-0.30 \pm 0.02$ b | $27.45 \pm 0.49$ a |
| | N3 | 0.956 | $-0.35 \pm 0.01$ b | $29.54 \pm 0.48$ a |
| | N0 | 0.945 | $-0.25 \pm 0.01$ a | $23.27 \pm 0.42$ b |
| S3 | N1 | 0.933 | $-0.28 \pm 0.01$ a | $26.05 \pm 0.53$ b |
| | N2 | 0.932 | $-0.35 \pm 0.02$ b | $31.14 \pm 0.68$ a |
| | N3 | 0.955 | $-0.37 \pm 0.00$ b | $34.01 \pm 0.38$ a |

### 3.2.3. Dynamics of the Grain Nitrogen Content under Different Treatments

As shown in Figure 6, the nitrogen content of Nanjing 9108 revealed no significant dynamics after anthesis.

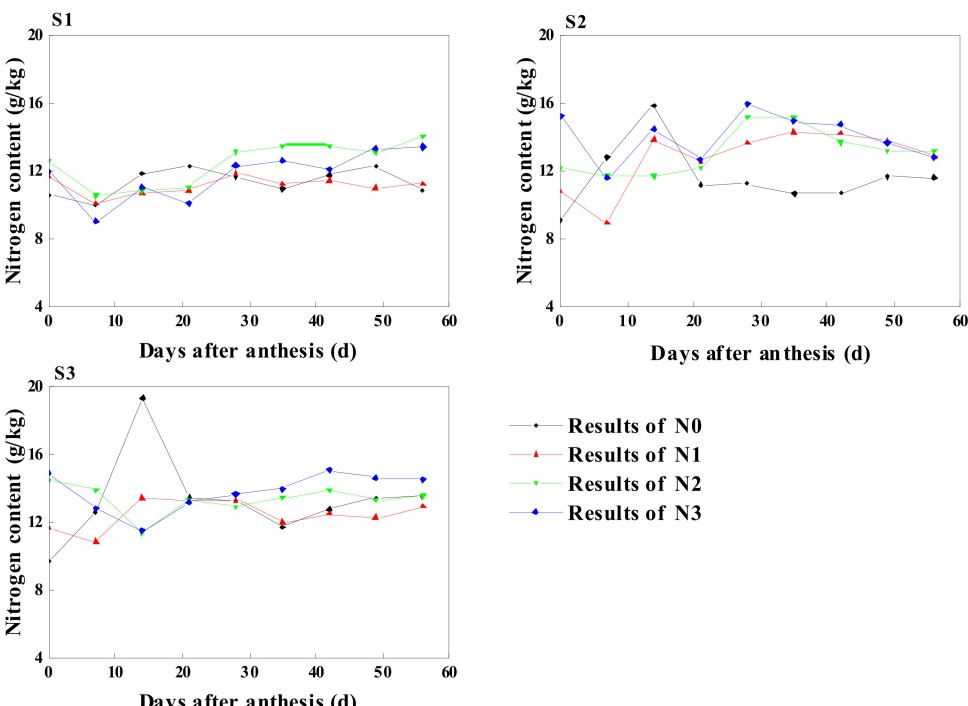

**Figure 6.** Change in nitrogen content of the grain (days after anthesis).

### 3.3. Dynamic Model and Characteristics of Dry Matter Accumulation in Rice Organs with Combination of Sowing Date and Nitrogen Application Rates

#### 3.3.1. Dynamics of Dry Matter Accumulation in the Stem Sheath under Different Treatments

The curve fitting results of the stem-sheath dry matter with days after anthesis are shown in Figure 7; the stem-sheath dry matter changed over time. The quadratic function $y = ax + bx^2 + c$ was used for quantitative description, and the meaning of each parameter was consistent with the nitrogen content of the stem sheath. The stem-sheath dry matter was the highest at the N3 level among the three sowing stages, but with the delay in the sowing date, the greater the influence of nitrogen on the stem-sheath dry matter and the greater the difference between the dry matter of the stem sheath at each nitrogen application level and the N0 level.

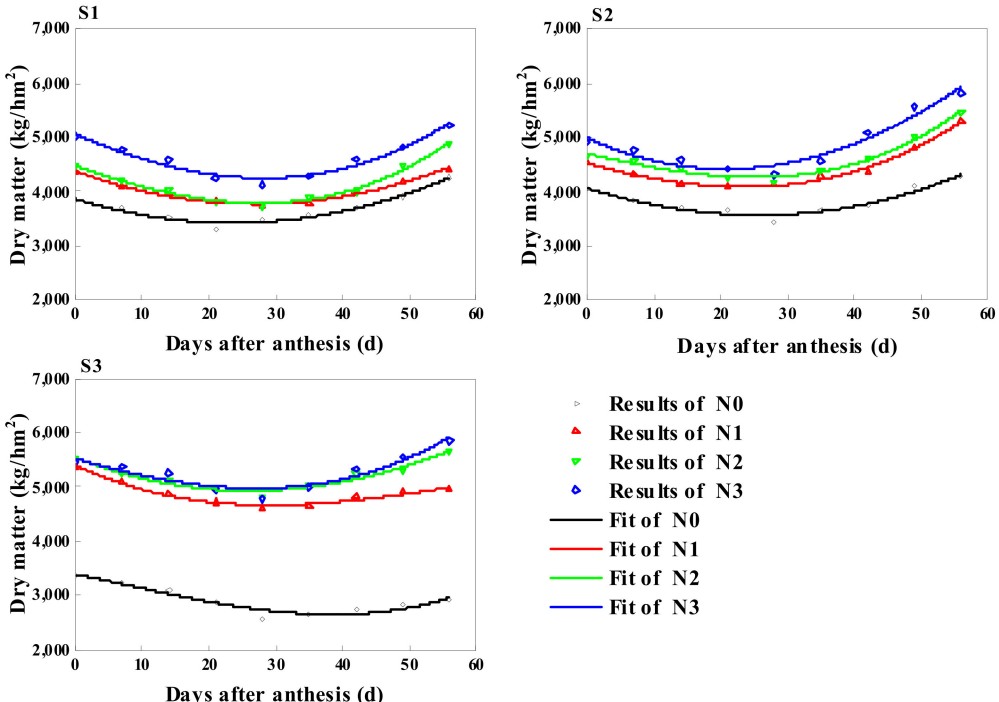

**Figure 7.** Change in dry matter of the stem- sheath (days after anthesis).

The dynamic change in the stem-sheath dry matter after anthesis is a quadratic function model. There were significant differences in the model parameters between the treatments, as shown in Table 4, and the coefficient of determination of each treatment model was between 0.743 and 0.984. The S1-treated $X_{sdm}$ had the lowest value at the N0 level, whereas the S2 and S3 treatments reached the maximum value at the N0 level; the lower the minimum point, the longer the stem-sheath dry matter decreased. $Y_{sdm}$ gradually increased with the increase in nitrogen application; except for the N0 level, the delay in other nitrogen levels with sowing date showed an upward trend—the greater the difference between the dry matter and minimum spread at the spike stage, the greater the decrease in the stem-sheath dry matter accumulation. The decrease in the stem-sheath dry matter accumulation in the S3 treatment was greater than that of S1 and S2, and more dry matter was transferred to the reproductive organs.

**Table 4.** Characteristic parameters of the fitted equation for the stem-sheath dry matter ($y = ax + bx^2 + c$). These values represent the arithmetic mean (average) of two duplicate values, with $\pm$ representing the standard error. Lowercase letters indicate significant differences (*p*-value < 0.05) in the stem-sheath dry matter at different nitrogen application rates within each sowing date.

| Treatment | | | | | | | |
|---|---|---|---|---|---|---|---|
| Sowing Dates | Nitrogen Levels | $R^2$ | $a_{sdm}$ | $b_{sdm}$ | $c_{sdm}$ | $X_{sdm}$ | $Y_{sdm}$ |
| S1 | N0 | 0.825 | −29.96 ± 1.88 a | 0.72 ± 0.03 b | 3842.9 ± 28.2 c | 20.83 ± 0.45 b | 3530.6 ± 2.1 c |
| | N1 | 0.964 | −44.43 ± 0.01 b | 0.82 ± 0.02 b | 4380.0 ± 9.6 b | 27.23 ± 0.81 a | 3775.0 ± 8.3 b |
| | N2 | 0.969 | −58.10 ± 0.04 c | 1.14 ± 0.06 a | 4524.9 ± 46.2 b | 25.47 ± 1.30 a | 3785.1 ± 8.1 b |
| | N3 | 0.921 | −62.49 ± 1.46 c | 1.16 ± 0.03 a | 5091.0 ± 9.5 a | 26.90 ± 0.01 a | 4250.5 ± 10.0 a |
| S2 | N0 | 0.907 | −39.78 ± 0.04 a | 0.78 ± 0.05 c | 4072.4 ± 38.5 d | 25.46 ± 1.63 a | 3566.0 ± 5.6 d |
| | N1 | 0.984 | −44.41 ± 0.31 b | 1.02 ± 0.03 b | 4557.2 ± 11.0 c | 21.89 ± 0.45 a | 4071.1 ± 4.3 c |
| | N2 | 0.954 | −46.75 ± 0.43 c | 1.05 ± 0.03 b | 4762.7 ± 12.4 b | 22.33 ± 0.43 a | 4240.9 ± 17.5 b |
| | N3 | 0.943 | −52.14 ± 0.41 d | 1.24 ± 0.00 a | 4976.4 ± 38.3 a | 21.07 ± 0.09 a | 4427.3 ± 31.8 a |
| S3 | N0 | 0.890 | −34.74 ± 0.37 a | 0.47 ± 0.01 d | 3430.1 ± 27.6 c | 37.33 ± 0.62 a | 2782.1 ± 31.5 c |
| | N1 | 0.743 | −41.06 ± 0.76 b | 0.65 ± 0.01 c | 5303.4 ± 58.3 b | 31.48 ± 0.24 b | 4657.2 ± 41.6 b |
| | N2 | 0.901 | −44.77 ± 0.57 c | 0.84 ± 0.02 b | 5529.5 ± 22.7 a | 26.52 ± 0.35 c | 4936.1 ± 22.4 a |
| | N3 | 0.875 | −47.36 ± 0.06 c | 0.95 ± 0.01 a | 5562.9 ± 5.05 a | 25.05 ± 0.33 c | 4969.8 ± 2.0 a |

### 3.3.2. Dynamics of the Leaf Dry Matter Accumulation under Different Treatments

The curve fitting results of the leaf dry matter with days after anthesis are shown in Figure 8. The leaf dry matter changed over time; the linear function $y = ax + b$ was used for fitting, and the parameters were consistent with the nitrogen content of the leaves. Except at the N0 level, the leaf dry matter of each nitrogen application level increased gradually with the delay in sowing date. Except for S3 treatment, each treatment increased with the increase in nitrogen application rate.

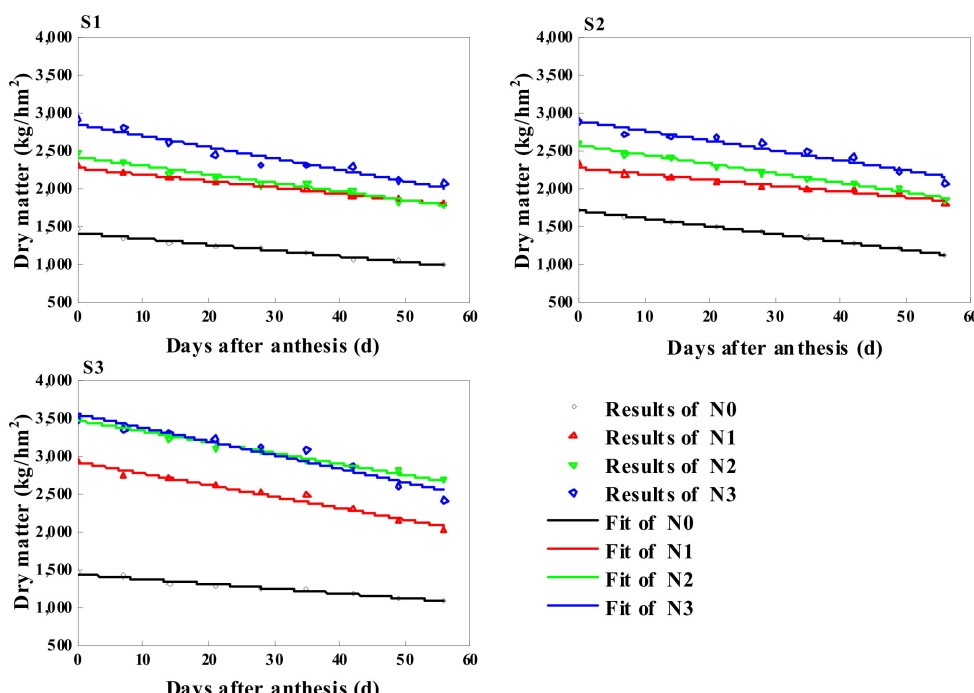

**Figure 8.** Change in dry matter of the leaves (days after anthesis).

As shown in Table 5, the main parameters of the dynamic change in the leaf dry matter after anthesis were significantly different between treatments, and the coefficient of

determination of each treatment curve was between 0.764 and 0.987. With the increase in nitrogen application, the absolute value of $a_{ldm}$ showed an upward trend, $b_{ldm}$ gradually increased, and under the interaction between the sowing date and nitrogen application, the absolute value and $b_{ldm}$ of the S3 treatment were greater and the maximum value was obtained at the N3 level. The greater the slope, the faster the leaf dry matter decreased, indicating that late sowing was conducive to the transfer of leaf dry matter.

**Table 5.** Characteristic parameters of the fitted equation for the leaf dry matter (y = ax + b). These values represent the arithmetic mean (average) of two duplicate values, with $\pm$ representing the standard error. Lowercase letters indicate significant differences (*p*-value < 0.05) in the leaf dry matter at different nitrogen application rates within each sowing date.

| Treatment | | $R^2$ | $a_{ldm}$ | $b_{ldm}$ |
|---|---|---|---|---|
| Sowing Dates | Nitrogen Levels | | | |
| S1 | N0 | 0.937 | $-7.67 \pm 0.21$ a | $1427.7 \pm 29.5$ c |
| | N1 | 0.955 | $-8.44 \pm 0.33$ ab | $2280.4 \pm 57.5$ b |
| | N2 | 0.935 | $-11.24 \pm 0.60$ b | $2425.2 \pm 14.5$ b |
| | N3 | 0.921 | $-14.88 \pm 0.92$ c | $2855.0 \pm 40.8$ a |
| S2 | N0 | 0.987 | $-10.34 \pm 0.06$ ab | $1723.0 \pm 238.0$ b |
| | N1 | 0.893 | $-7.61 \pm 1.23$ a | $2279.3 \pm 151.2$ ab |
| | N2 | 0.979 | $-12.34 \pm 0.52$ b | $2583.2 \pm 63.1$ ab |
| | N3 | 0.764 | $-17.42 \pm 0.42$ c | $2974.8 \pm 88.8$ a |
| S3 | N0 | 0.957 | $-6.23 \pm 0.73$ a | $1434.0 \pm 44.0$ c |
| | N1 | 0.763 | $-16.13 \pm 0.55$ b | $2975.4 \pm 80.2$ b |
| | N2 | 0.955 | $-14.24 \pm 0.65$ b | $3459.9 \pm 62.3$ ab |
| | N3 | 0.882 | $-22.91 \pm 1.09$ c | $3543.0 \pm 156.9$ a |

### 3.3.3. Dynamics of the Grain Dry Matter Accumulation under Different Treatments

The change curve of the dry matter under different treatments with days after anthesis is shown in Figure 9. The grain dry matter changed over time, showing an S-shaped upward slow–fast–slow trend, which is in line with the characteristics of the logistic growth curve; therefore, Formula (1) was used to fit the dry matter accumulation and change in grain. Except for the N0 levels of S1 and S3, the difference between the nitrogen application levels was not significant, and the dry matter of the grain under the S2 treatment was N2 > N3 > N1 > N0.

The main parameters of the dynamic change in the grain dry matter accumulation are shown in Table 6. The coefficient of determination of each treatment curve was between 0.833 and 0.995, indicating that this growth curve equation accurately described the change process of rice grain dry matter accumulation with the growth time after anthesis. The theoretical maximum accumulation of dry matter ($k_{gdm}$, kg/hm$^2$), the start and end time ($t1_{gdm}$, d), ($t2_{gdm}$, d), and the time to reach the maximum accumulation rate ($T_{gdm}$, d) of the grains under different treatments were different. With the increase in nitrogen fertilizer, $k_{gdm}$ showed a trend of first increasing and then decreasing in the S1 and S2 treatments and gradually increased in the S3 treatment; the maximum value of 9652.7 kg/hm$^2$ was obtained in the S3N3 treatment. $t2_{gdm} - t1_{gdm}$ was the rapid grain filling stage; except for the N0 levels, the $T2_{gdm} - t1_{gdm}$ treated with S1 and S3 gradually decreased with the increase in nitrogen levels. The $T_{gdm}$ in the S1 and S3 treatments obtained the lowest value at the N2 level, but the S2 treatment obtained the maximum value at N2. With the interaction of sowing date and nitrogen application rate, the rapid grouting period of the S1 treatment lasted the longest, at 32 days at the N0 level.

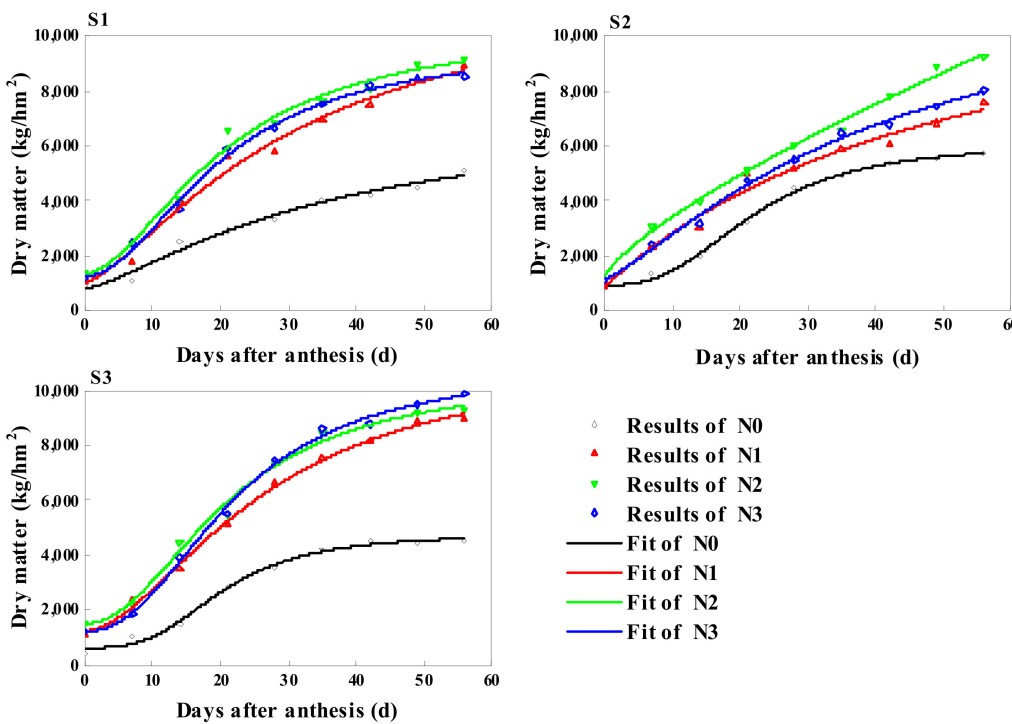

**Figure 9.** Change in dry matter of the grain (days after anthesis).

**Table 6.** Characteristic parameters of the fitted equation for the grain dry matter ($y = k/(1 + ae^{-bx})$). These values represent the arithmetic mean (average) of two duplicate values, with $\pm$ representing the standard error. Lowercase letters indicate significant differences ($p$-value < 0.05) in the grain dry matter at different nitrogen application rates within each sowing date.

| Treatment | | | | | | | |
|---|---|---|---|---|---|---|---|
| Sowing Dates | Nitrogen Levels | $R^2$ | $k_{gdm}$ | $a_{gdm}$ | $b_{gdm}$ | $t2_{gdm} - t1_{gdm}$ | $T_{gdm}$ |
| S1 | N0 | 0.972 | 5070.3 $\pm$ 20.2 c | 4.25 $\pm$ 0.02 b | 0.08 $\pm$ 0.00 a | 32.92 $\pm$ 0.00 a | 18.08 $\pm$ 0.05 a |
| | N1 | 0.960 | 8263.9 $\pm$ 89.6 b | 5.17 $\pm$ 0.11 ab | 0.10 $\pm$ 0.00 a | 26.75 $\pm$ 0.14 ab | 16.68 $\pm$ 0.31 a |
| | N2 | 0.982 | 8986.4 $\pm$ 71.5 a | 5.01 $\pm$ 0.01 ab | 0.11 $\pm$ 0.00 a | 24.97 $\pm$ 0.12 ab | 15.27 $\pm$ 0.09 a |
| | N3 | 0.940 | 8248.8 $\pm$ 97.6 b | 7.05 $\pm$ 0.88 a | 0.14 $\pm$ 0.03 a | 19.65 $\pm$ 3.88 b | 14.33 $\pm$ 1.93 a |
| S2 | N0 | 0.975 | 5740.3 $\pm$ 252.6 c | 7.98 $\pm$ 0.17 a | 0.12 $\pm$ 0.01 b | 23.11 $\pm$ 1.52 a | 18.23 $\pm$ 1.38 a |
| | N1 | 0.833 | 6477.9 $\pm$ 41.2 bc | 6.06 $\pm$ 0.22 b | 0.14 $\pm$ 0.01 ab | 18.62 $\pm$ 1.18 ab | 12.72 $\pm$ 0.56 b |
| | N2 | 0.900 | 7996.2 $\pm$ 58.8 a | 4.30 $\pm$ 0.01 c | 0.12 $\pm$ 0.00 ab | 22.42 $\pm$ 0.10 ab | 12.41 $\pm$ 0.07 b |
| | N3 | 0.903 | 7202.5 $\pm$ 19.8 b | 6.89 $\pm$ 0.49 ab | 0.16 $\pm$ 0.01 a | 17.19 $\pm$ 0.62 b | 12.55 $\pm$ 0.01 b |
| S3 | N0 | 0.995 | 4609.5 $\pm$ 202.2 b | 10.30 $\pm$ 1.58 a | 0.13 $\pm$ 0.02 a | 20.70 $\pm$ 2.41 a | 18.09 $\pm$ 0.91 a |
| | N1 | 0.988 | 9024.4 $\pm$ 81.6 a | 6.12 $\pm$ 0.40 a | 0.10 $\pm$ 0.00 a | 26.62 $\pm$ 0.54 a | 18.27 $\pm$ 0.28 a |
| | N2 | 0.984 | 9420.7 $\pm$ 61.1 a | 5.82 $\pm$ 0.83 a | 0.11 $\pm$ 0.01 a | 25.08 $\pm$ 2.37 a | 16.54 $\pm$ 0.21 a |
| | N3 | 0.985 | 9652.7 $\pm$ 21.1 a | 9.20 $\pm$ 0.77 a | 0.13 $\pm$ 0.01 a | 20.57 $\pm$ 1.20 a | 17.27 $\pm$ 0.36 a |

*3.4. Dynamic Model of Nitrogen Uptake of the Rice Organs Combined with Sowing Date and Nitrogen Application Rate*

3.4.1. Dynamics of the Stem-Sheath Nitrogen Uptake under Different Treatments

The change curve of the stem-sheath nitrogen uptake with days after anthesis is shown in Figure 10. The stem-sheath nitrogen content changed over time, which was the same as that of the stem-sheath dry matter and nitrogen content, and was fitted by the quadratic function $y = ax + bx^2 + c$. The nitrogen uptake of the stem sheath in the three sowing dates was N3 > N2 > N1 > N0, but the S3-treated stem-sheath nitrogen uptake rate, in addition to the N0 levels, was higher than that of S1 and S2.

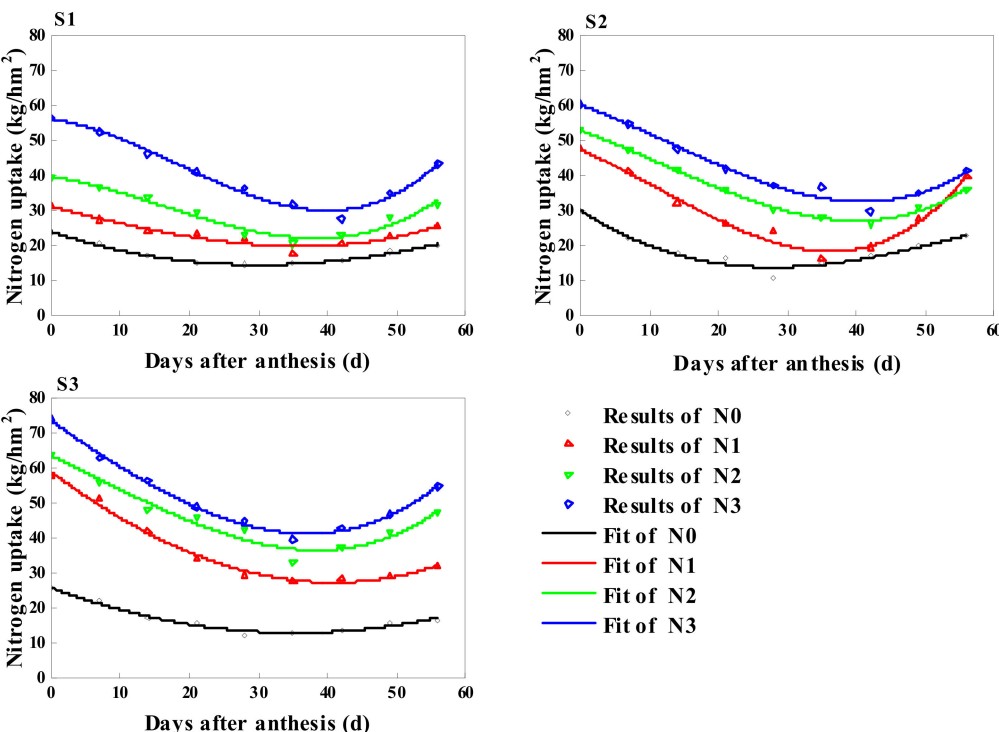

**Figure 10.** Change in nitrogen uptake of the stem-sheath (days after anthesis).

The main parameters of the dynamic change in the stem-sheath nitrogen uptake after anthesis were significantly different between the treatments, as shown in Table 7. The coefficient of determination of each treatment curve was between 0.861 and 0.978. $X_{snu}$, except for the S3 treatment, gradually increased with the increase in nitrogen fertilizer, and the postponement of the sowing date showed an upward trend except for the N1 level. $Y_{snu}$ showed an upward trend with the increase in nitrogen application; except for the N0 level, the postponement of the sowing date gradually increased. Under the interaction of the sowing date and nitrogen application, the maximum value of 42.26 kg/hm² was obtained in the S3N3 treatment.

**Table 7.** Characteristic parameters of the fitted equation for the stem-sheath nitrogen uptake ($y = ax + bx^2 + c$). These values represent the arithmetic mean (average) of two duplicate values, with $\pm$ representing the standard error. Lowercase letters indicate significant differences (*p*-value < 0.05) in the stem-sheath nitrogen uptake at different nitrogen application rates within each sowing date.

| Treatment | | $R^2$ | $a_{snu}$ | $b_{lnu}$ | $c_{lnu}$ | $X_{lnu}$ | $Y_{lnu}$ |
|---|---|---|---|---|---|---|---|
| Sowing Dates | Nitrogen Levels | | | | | | |
| S1 | N0 | 0.963 | −0.60 ± 0.05 a | 0.010 ± 0.001 b | 24.00 ± 1.40 d | 30.11 ± 0.56 b | 14.96 ± 0.83 a |
| | N1 | 0.910 | −0.68 ± 0.03 a | 0.010 ± 0.000 b | 31.85 ± 0.44 c | 33.70 ± 1.45 ab | 20.47 ± 0.55 b |
| | N2 | 0.861 | −1.04 ± 0.02 b | 0.015 ± 0.000 a | 42.29 ± 0.54 b | 34.72 ± 0.62 ab | 24.20 ± 0.11 c |
| | N3 | 0.886 | −1.41 ± 0.06 c | 0.019 ± 0.001 a | 59.94 ± 0.95 a | 37.14 ± 0.34 a | 33.76 ± 0.05 a |
| S2 | N0 | 0.911 | −1.00 ± 0.01 a | 0.017 ± 0.001 b | 29.30 ± 0.37 c | 30.27 ± 0.51 b | 14.19 ± 0.42 d |
| | N1 | 0.897 | −1.87 ± 0.02 c | 0.029 ± 0.001 a | 50.86 ± 2.26 b | 32.27 ± 1.40 b | 20.70 ± 0.69 c |
| | N2 | 0.959 | −1.35 ± 0.01 b | 0.018 ± 0.001 b | 55.08 ± 0.06 b | 38.54 ± 0.85 a | 29.11 ± 0.46 b |
| | N3 | 0.936 | −1.39 ± 0.03 b | 0.018 ± 0.001 b | 62.46 ± 0.15 a | 39.55 ± 0.36 a | 35.09 ± 0.44 a |
| S3 | N0 | 0.975 | −0.73 ± 0.03 a | 0.011 ± 0.001 c | 25.59 ± 0.37 d | 34.53 ± 0.67 a | 13.09 ± 0.48 c |
| | N1 | 0.978 | −1.55 ± 0.01 b | 0.019 ± 0.000 b | 58.84 ± 1.06 c | 40.67 ± 0.20 a | 27.42 ± 1.37 b |
| | N2 | 0.922 | −1.49 ± 0.04 b | 0.020 ± 0.002 ab | 64.94 ± 0.85 b | 37.45 ± 2.70 a | 37.15 ± 2.07 a |
| | N3 | 0.975 | −1.84 ± 0.05 c | 0.026 ± 0.001 a | 74.57 ± 0.54 a | 35.27 ± 0.49 a | 42.26 ± 0.19 a |

### 3.4.2. Dynamics of the Leaf Nitrogen Uptake under Different Treatments

The change curve of the leaf nitrogen uptake with days after anthesis is shown in Figure 11. The trend of leaf nitrogen uptake over time was the same as that of the leaf nitrogen content. The decreasing trend was exponential, the main characteristic parameters of the dynamic change in nitrogen uptake of each treated leaf were calculated by the index function $y = ae^{bx}$, and the meaning of each characteristic parameter was consistent with the leaf nitrogen content. Overall, leaf nitrogen uptake increased with the delay in sowing date and gradually increased with the increase in nitrogen application.

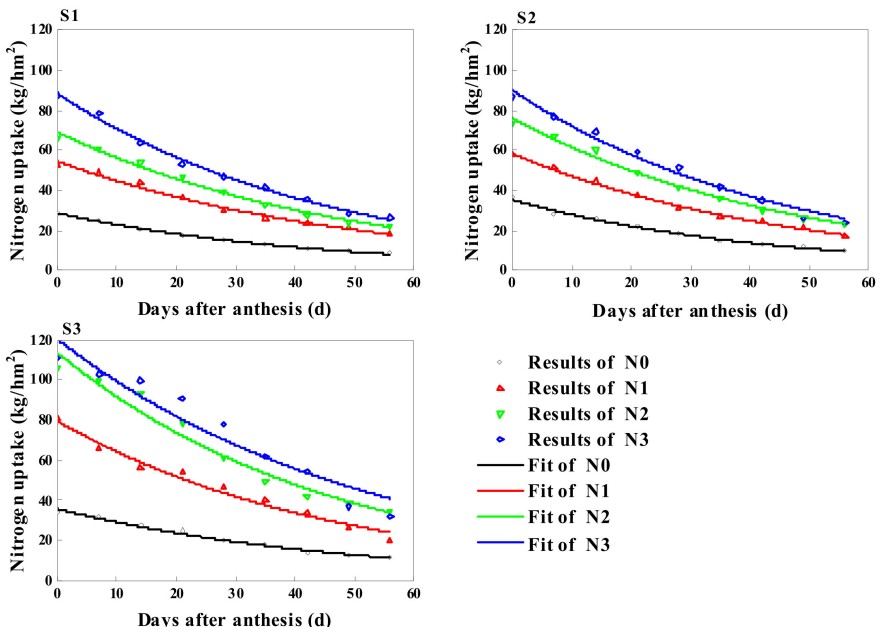

**Figure 11.** Change in nitrogen uptake of the leaves (days after anthesis).

The main parameters of the dynamic change in the leaf nitrogen uptake after anthesis are shown in Table 8, and the coefficient of determination of each treatment curve was between 0.930 and 0.987. There were significant differences in $a_{lnu}$ between the treatments, and the maximum value of $b_{lnu}$ was 129.97 kg/hm$^2$ in the S3N3 treatment; there was no significant difference in $b_{lnu}$ with the delay in the sowing date and the increase in nitrogen.

**Table 8.** Characteristic parameters of the fitted equation for the leaf nitrogen uptake ($y = ae^{bx}$). These values represent the arithmetic mean (average) of two duplicate values, with $\pm$ representing the standard error. Lowercase letters indicate significant differences (*p*-value < 0.05) in the leaf nitrogen uptake at different nitrogen application rates within each sowing date.

| Treatment | | $R^2$ | $a_{lnu}$ | $b_{lnu}$ |
|---|---|---|---|---|
| Sowing Dates | Nitrogen Levels | | | |
| | N0 | 0.977 | 28.13 ± 0.55 d | −0.02 ± 0.00 a |
| S1 | N1 | 0.968 | 53.83 ± 0.86 c | −0.02 ± 0.00 a |
| | N2 | 0.971 | 70.14 ± 0.47 b | −0.02 ± 0.00 a |
| | N3 | 0.986 | 87.67 ± 2.05 a | −0.02 ± 0.00 a |
| | N0 | 0.987 | 34.96 ± 4.11 c | −0.02 ± 0.00 a |
| S2 | N1 | 0.958 | 57.58 ± 3.16 b | −0.02 ± 0.00 a |
| | N2 | 0.974 | 74.19 ± 0.31 ab | −0.02 ± 0.00 a |
| | N3 | 0.968 | 94.56 ± 5.07 a | −0.03 ± 0.00 aa |
| | N0 | 0.963 | 35.85 ± 2.74 c | −0.02 ± 0.00 a |
| S3 | N1 | 0.937 | 82.17 ± 0.22 b | −0.02 ± 0.00 a |
| | N2 | 0.966 | 113.41 ± 4.36 a | −0.02 ± 0.00 a |
| | N3 | 0.930 | 129.97 ± 4.98 a | −0.02 ± 0.00 a |

### 3.4.3. Dynamics of the Grain Nitrogen Uptake under Different Treatments

The change curve of the grain nitrogen uptake with days after anthesis is shown in Figure 12. The grain nitrogen uptake rate changed over time along with the grain dry matter, showing a slow–fast–slow S-shaped upward trend. Formula (1) was used to fit the grain dry matter accumulation change process, and the S1 and S2 treatments of grain nitrogen uptake were N2 > N3 > N1 > N0, whereas the S3 treatment gradually increased with the increase in nitrogen level.

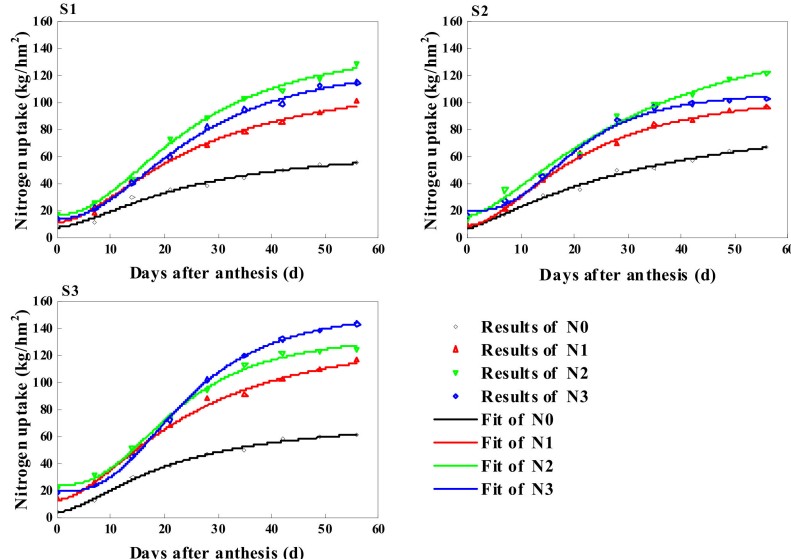

**Figure 12.** Change in nitrogen uptake of the grain (days after anthesis).

The main parameters of the dynamic change in the grain nitrogen uptake are shown in Table 9. The coefficient of determination of each treatment curve exceeded 0.964. The theoretical maximum nitrogen uptake of the grain ($k_{gnu}$, kg/hm$^2$), the start and end time of rapid nitrogen absorption into the grain ($t1_{gnu}$, d), ($t2_{gnu}$, d), and the time to reach the maximum absorption rate ($T_{gnu}$, d) were significantly different under the different treatments. For $k_{gnu}$, in addition to the N0 levels, the average nitrogen levels were S3 > S1 > S2, and with the increase in nitrogen fertilizer, the trend of first increasing and then decreasing in the S1 and S2 treatments was shown, and the maximum value was obtained at the N2 level, whereas S3 gradually increased. The earliest seeding period to enter the rapid absorption of grain nitrogen was S1, followed by S2. $t2_{gnu} - t1_{gnu}$ was the time when grain absorbed nitrogen rapidly, and the sowing dates were ranked as S1 > S2 > S3. Under the interaction between the sowing date and nitrogen application, the longest duration of rapid nitrogen absorption in grains occurred in the S1N0 treatment, indicating that different nitrogen levels had different effects on different sowing dates. With the increase in nitrogen application, the time for the S1 and S3 treatments to enter the maximum nitrogen absorption rate of the grain was gradually extended.

**Table 9.** Characteristic parameters of the fitted equation for the grain nitrogen uptake ($y = k/(1 + ae^{-bx})$). These values represent the arithmetic mean (average) of two duplicate values, with $\pm$ representing the standard error. Lowercase letters indicate significant differences (*p*-value < 0.05) in the stem-sheath nitrogen uptake at different nitrogen application rates within each sowing date.

| Treatment | | $R^2$ | $k_{gnu}$ | $a_{gnu}$ | $b_{gnu}$ | $t2_{gnu} - t1_{gnu}$ | $T_{gnu}$ |
|---|---|---|---|---|---|---|---|
| Sowing Dates | Nitrogen Levels | | | | | | |
| S1 | N0 | 0.966 | 56.28 ± 0.51 c | 4.56 ± 0.03 c | 0.09 ± 0.00 a | 28.95 ± 0.32 a | 16.67 ± 0.25 b |
| | N1 | 0.971 | 97.39 ± 0.93 b | 5.71 ± 0.13 b | 0.10 ± 0.00 a | 27.73 ± 0.00 a | 18.33 ± 0.24 ab |
| | N2 | 0.982 | 128.16 ± 5.70 a | 6.05 ± 0.00 b | 0.09 ± 0.00 a | 29.13 ± 0.81 a | 19.91 ± 0.55 a |
| | N3 | 0.964 | 115.13 ± 2.43 ab | 7.13 ± 0.07 a | 0.10 ± 0.00 a | 27.04 ± 0.70 a | 20.16 ± 0.61 a |
| S2 | N0 | 0.983 | 63.76 ± 0.17 d | 5.28 ± 0.15 a | 0.10 ± 0.00 a | 25.95 ± 0.13 a | 16.39 ± 0.20 a |
| | N1 | 0.975 | 92.25 ± 0.22 c | 6.85 ± 0.68 a | 0.12 ± 0.01 a | 22.26 ± 1.68 a | 16.15 ± 0.39 a |
| | N2 | 0.984 | 120.83 ± 1.94 a | 5.57 ± 0.30 a | 0.09 ± 0.00 a | 28.48 ± 0.46 a | 18.57 ± 0.89 a |
| | N3 | 0.978 | 104.15 ± 1.52 b | 6.35 ± 0.56 a | 0.12 ± 0.01 a | 22.93 ± 1.69 a | 16.00 ± 0.42 a |
| S3 | N0 | 0.986 | 58.43 ± 0.31 d | 7.30 ± 0.09 b | 0.13 ± 0.00 a | 20.99 ± 0.09 a | 15.84 ± 0.16 b |
| | N1 | 0.988 | 113.37 ± 2.19 c | 6.34 ± 0.01 c | 0.11 ± 0.01 a | 25.62 ± 1.00 a | 17.96 ± 0.71 ab |
| | N2 | 0.981 | 125.48 ± 2.41 b | 7.17 ± 0.21 b | 0.12 ± 0.01 a | 22.11 ± 1.21 a | 16.52 ± 0.66 ab |
| | N3 | 0.996 | 143.69 ± 0.56 a | 9.92 ± 0.14 a | 0.11 ± 0.01 a | 23.83 ± 1.51 a | 20.75 ± 1.19 a |

## 4. Discussion

Dry matter accumulation is the material basis for yield formation during crop growth, and the dry matter accumulation of rice under different treatments differs; quantitative analysis of the dynamic changes in dry matter accumulation during crop growth is of great significance to reveal crop yield formation [33]. Nitrogen uptake in rice varies with growth, and models can quickly and easily estimate nitrogen needs for rice growth [28]. In this experiment, the dry matter accumulation and nitrogen accumulation processes of each organ of different rice plants were fitted by different models, and the relevant characteristic parameters in the fitting equation were calculated. In addition, the dynamic characteristics of dry matter accumulation and nitrogen uptake and allocation processes were quantitatively analyzed.

The SPAD value was significantly correlated with the leaf nitrogen content, which was a sensitive indicator reflecting the dynamic change in nitrogen of plants; the nitrogen status during rice growth was monitored; and the nitrogen application was guided by nitrogen application to obtain high yields and improve nitrogen use efficiency [34–36].The results of this study show that the change trend in the SPAD values of the upper three leaves after anthesis was an inverted S-shaped downward curve, which was basically consistent with the study of Zhao et al. [37]. In contrast, some studies found that the SPAD values of the upper four leaves after anthesis of rice linearly decreased [38], which is different from the results of this study and may be due to significant differences in the nitrogen application rate or the varieties or settings selected for the experiment. The results of this study show that the duration of the maximum SPAD value ($k_s$) and rapid descent period ($t2_s - t1_s$) in each of the upper three leaves increased with the increase in nitrogen fertilizer, indicating that the more nitrogen applied, the longer and slower the rapid decline of nitrogen in leaves. This is the same as the result of the gradual increase in the SPAD value of the leaves with the increase in nitrogen application in Huang et al. [39]. In this study, the leaf nitrogen content and dry matter showed a linear downward trend, and the nitrogen uptake rate showed an exponential downward trend, similar to the results reported by Xu et al. [40]. Cao et al. [41] conducted four studies that showed that the change trend in nitrogen uptake of post-anthesis grains at different sowing periods and nitrogen application rates could be dynamically fitted using the NRMSE model, and the fitting effect was good. The change trend in the dry matter and nitrogen uptake in the grain in this study was an S-type upward curve, and the fitting effect of the inverted logistic model was excellent. That is inconsistent

with previous research results and may be due to certain differences in external factors such as the experimental environment and cultivation measures. The change trend in dry matter and nitrogen uptake was a parabolic line with an upward-pointing opening, which is basically consistent with the results of Xu [42].

The dry matter and nitrogen uptake of the stem sheath and leaves of the late sowing were higher than those of other treatments, and the dry matter weight and nitrogen uptake of the grain entered the rapid decline rate ($T_{lnu}$) the earliest. This is inconsistent with Sun et al.'s [43] study in that the contribution rate of the stem sheath and leaf dry matter to the panicle and the contribution rate of the stem sheath and leaf nitrogen to the panicle were all higher than the suitable sowing date. Pal et al. [44] found that the source activity of late-sown rice during the reproductive growth was limited, resulting in low dry matter accumulation and nitrogen uptake during flowering, and that increased dry matter accumulation after anthesis, coupled with increased nitrogen uptake and transport during flowering, led to an increase in reservoir capacity. However, in this experiment, the dry matter accumulation and nitrogen uptake of rice after late sowing were higher than those of other treatments, and they increased with the increase in nitrogen fertilizer. The research results are inconsistent with previous studies, which were likely to have been affected by the interaction between rice varieties, regional environment, sowing date, and nitrogen application rate. The S1 and S2 treatments had higher nitrogen uptake at the N2 level and the S3 treatment had higher nitrogen uptake at the N3 level. This was consistent with the results of Stone et al.'s [45] study in that late sowing could reduce nitrogen loss, indicating that late sowing could increase the source capacity and reservoir capacity at high nitrogen levels and promote the absorption and accumulation of nitrogen by plants. This study fills the gap in this regard by making up for the lack of attention to nitrogen dynamics in the aboveground plant parts except after wheat anthesis [46,47]. There are few reports on the nitrogen dynamics after flowering in rice; hence, this study lays a foundation for the study of nitrogen dynamics after anthesis.

## 5. Conclusions

In this experiment, the delay in sowing date and the increase in nitrogen application were conducive to the dry matter accumulation and nitrogen uptake and utilization of the stem sheath and leaf of Nanjing 9108. This accelerated the transfer of leaf dry matter and nitrogen to reproductive organs and promoted the accumulation of grain nitrogen; dry matter accumulation and nitrogen absorption were faster under the N2 level of the S1 and S2 treatments. However, the absorption and accumulation were greater under the N3 level of the S3 treatment. The late-sowing rice was more conducive to enhanced leaf photosynthesis, increased dry matter accumulation, the promotion of grain grouting, and improved nitrogen use efficiency at high nitrogen levels.

**Author Contributions:** Y.Y.: Conceived and designed the experiments, performed the experiments, analyzed the data, prepared figures and/or tables, authored or reviewed drafts of the paper, and approved the final draft. K.Z.: Performed experiments and helped with data analysis. J.M.: Performed experiments and helped with data analysis. L.H.: Conceived the study, participated in its design and coordination, helped to revise the manuscript, and approved the final draft. H.Z.: Conceived the study, participated in its design and coordination, helped to revise the manuscript, and approved the final draft. Y.Y., K.Z. and J.M. contributed equally to this work and should be considered co-first authors. All authors have read and agreed to the published version of the manuscript.

**Funding:** This research was funded by the National Key Research and Development Program of China, grant number 31571596.

**Institutional Review Board Statement:** Not applicable.

**Informed Consent Statement:** Not applicable.

**Data Availability Statement:** Not applicable.

**Conflicts of Interest:** The authors declare no conflict of interest.

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
