# Peer review of "Post-Anthesis Nitrogen Dynamic Models and Characteristics of Rice Combined with Sowing Date and Nitrogen Application Rate"

_sustainability, doi:10.3390/su14094956_

Round 1
Reviewer 1 Report
The manuscript needs thorough editing and rewriting in most part. The Abstract doesn't convey the clear message. It needs rework. Lot of grammatical errors that needs correction as indicated. The data is too elaborate and can be made succinct. The figures are unclear and with errors in text. Overall it needs intense revision to improve its quality.

Author Response
Response to Reviewer Comments
Dear reviewer,
Thank you for reviewing this manuscript in your busy schedule. Now we have carefully corrected and replied the manuscript for this revision. The language and grammar in the manuscript have been improved, and the clarity of all figures have also been processed. The revision instructions are as follows:
Point 1: Too long sentence, doesn't have clarity:To explore effects of sowing date and nitrogen application rate combined with post-anthesis nitrogen uptake and distribution of superior japonica rice cultivar Nanjing 9108, and to investigate the interrelationship between nitrogen status of plants at different sowing dates and post-anthesis nitrogen uptake and translocation, three sowing dates were scheduled on May 23rd (S1), June 2nd (S2), and June 12 (S3), and four treatments were applied including no nitrogen (N0), 180 (N1), 270 (N2), and 360 (N3) kg N∙hm-2 for field experiments. Curve fitting methods were performed for quantitative analysis of the dynamic characteristics of nitrogen uptake and utilization process of treated rice after full heading.
Response 1: This section has been modified in the manuscript, please see P1, L 13-18.
Point 2: The meaning of the sentence is not clear: However, the maximum SPAD value, starting and ending time of rapid decline, and decline rate differed from various treatments.
Response 2: The sentence has been changed, and as follows: But the maximum SPAD value (ks), the time to enter the rapid period of decline (t1s), and the time to reach the maximum rate (Ts) were different between the different treatments.
Point 3: Again, very long sentence and doesn't convey the meaning clearly: The nitrogen content and dry mater weight of leaves decreased linearly among different treatments. Leaf nitrogen uptake exhibited an exponential downward trend, and the rate of decline and theoretical nitrogen content at full heading stage, dry matter accumulation and nitrogen uptake elevated gradually with the delay of sowing date and the increase of nitrogen, and the maximum value occurred after S3N3 treatment.
Response 3: The sentence has been changed, and as follows: The leaf nitrogen content and dry matter of different treatments exhibited a linear curve, the leaf nitrogen uptake rate showed an exponential downward trend, and the decline rate and the nitrogen content, dry matter accumulation, and nitrogen uptake at the theoretical spike stage gradually increased with the delay of the sowing date and the increase in nitrogen; the maximum values were obtained in the S3N3 treatment.
Point 4: Entire abstract is unclear. Please rewrite everything in simple, short and clear sentences about what you want to convey.
Response 4: The abstract has been rewritten, please see P1, L 13-29.
Point 5: Confusing and unclear statement: The crop growth model can present basic laws and quantitative relationships of the crop growth and development process, which can also quantitatively predict the dynamic behavior and yield quality of the crop growth system, thereby assisting the optimal management and quantitative regulation of crop growth and yield systems, and achieving the objectives of high-yield, high-efficiency, ecological, high-quality, and safe crop production.
Response 5: The sentence has been changed, and as follows: A crop growth model can present the basic dynamics and quantitative relationships of crop growth and the development process, quantitatively predict the dynamic behavior and yield quality of crop growth systems, assist in optimizing management and quantitative regulation, and achieve the goals of high yield, high efficiency, eco-friendly, high quality, and safety of crop production.
Point 6: Long sentences: The present trial applied high-quality japonica rice cultivar Nanjing 9108 as materials, which has been widely planted in Jiangsu Province, to clarify the quantitative relationship between different sowing dates and different nitrogen application rates on post-anthesis nitrogen uptake and utilization of rice.
Response 6: The sentence has been changed, and as follows: In this experiment, carried out in Jiangsu Province, the high-quality rice variety Nanjing 9108 was used as the material. This experiment analyzed changes in the dynamic characteristics of nitrogen after anthesis of rice with different sowing dates and nitrogen application rates and their relationship with utilization efficiency.
Point 7: Unclear: The change trend of nitrogen content with time in stem and sheath was identical to the dry matter of stem and sheath, both of which presented as a parabola with an upward opening, and could be fitted using the quadratic function y=ax+bx2+c.
Response 7: The sentence has been changed, and as follows: The changes in the stem-sheath nitrogen content over time were the same as those of the stem-sheath dry matter. The quadratic function y=ax+bx2+c was used for fitting, and a, b, and c were the parameters to be determined.
Point 8: All dynamic changes of stem and sheath dry matter weight after full heading presented a quadratic function model with an opening upward. what is opening upward mean?
Response 8: The sentence has been deleted.
Point 9: Too long sentence: The change trend of nitrogen content with time in stem and sheath was identical to the dry matter of stem and sheath and nitrogen content, both of which presented as a parabola with an upward opening, and could be fitted using the quadratic function y=ax+bx2+c.
Response 9: The sentence has been changed, and as follows: The stem-sheath nitrogen content changed over time, which was the same as that of the stem-sheath dry matter and nitrogen content, and was fitted by the quadratic function y=ax+bx2+c.
Point 10: This should go in the material and methods section: A chlorophyll meter was utilized to monitor leaf N status of rice and the application of nitrogen fertilizer under guidance can assist in achieving high yield and improving nitrogen utilization efficiency.
Response 10: The sentence has been deleted.
Thank you for your consideration. I look forward to hearing from you.
Sincerely,
Ying Ye
March.28,2022

Reviewer 2 Report
In my opinion, the subject of this article falls within the area of ​​interest of MDPI Plants and that is where it should be addressed.Author Response
Response to Reviewer 2 Comments
Dear reviewer,
Thank you for reviewing this manuscript in your busy schedule. Now we have carefully corrected and replied the manuscript for this revision. The language and grammar in the manuscript have been improved, and the clarity of all figures have also been processed.
We do understand that you are not satisfied with the manuscript. However, we still want to modify the manuscript according to the comments of the other three reviewers and academic editors. We hope you can review our revised manuscript again. We really appreciate your comments, because your comments can make our research perfect.
We hope to receive your approval. We look forward to hearing from you.
Sincerely,
Ying Ye
Kaocheng Zhao
March.28,2022

Reviewer 3 Report
The topic of the manuscript falls within the general scope of the journal and contains potentially useful information for readers. It is acceptable for publication as its present form. I only emphasize that texts in figures are soo small and hard to read.
The more explanation are needed for determining the models.
The other parts of manuscript are well written and no need for revision.
Author Response
Response to Reviewer Comments
Dear reviewer,
Thank you for reviewing this manuscript in your busy schedule and your affirmation.
As you suggested, we have made corresponding modifications, as follows:
Comment
“The topic of the manuscript falls within the general scope of the journal and contains potentially useful information for readers. It is acceptable for publication as its present form. I only emphasize that texts in figures are soo small and hard to read.
The more explanation are needed for determining the models.
The other parts of manuscript are well written and no need for revision.”
Response:
Through analyzing your comments, we revise this paper, improve the language and grammar in the manuscript, and processe the clarity of all figures.
Thank you for your consideration. I look forward to hearing from you.
Sincerely,
Ying Ye
March.28,2022

Reviewer 4 Report
Sustainability-MDPI
MS No.: sustainability-1634069
Manuscript Title: "Post-anthesis nitrogen dynamic models and characteristics of rice combined with sowing date and nitrogen application rate"
Reviewer comments:
In this current study, the authors studied the impacts of sowing sate and nitrogen application rate on the post-anthesis nitrogen dynamic models of rice plants.
The topic is nice and fits well with the scope of Sustainability-MDPI, and the results are of interest to the scientific community. The study is well-designed and the methods are satisfactory. However, the text needs a major revision before publication. In addition, this paper gives an overview of the response of post-anthesis nitrogen dynamic models as a result of sowing date and nitrogen rates treatments. While the innovation is insufficient and some of the discussion is inadequate. It will be deserved a major revision before consideration of publication in Sustainability-MDPI.
- P1, L13-18, To explore effects of sowing date and nitrogen application rate combined with post-anthesis nitrogen uptake and distribution of superior japonica rice cultivar Nanjing 9108, and to investigate the interrelationship between nitrogen status of plants at different sowing dates and post-anthesis nitrogen uptake and translocation, three sowing dates were scheduled on May 23rd (S1), June 2nd (S2), and June 12 (S3), and four treatments were applied including no nitrogen (N0), 180 (N1), 270 (N2), and 360 (N3) kg N∙hm-2 for field experiments. One sentence consists of seven lines, the sentence is too long, please shorten it and be specific.
- P1, L 18-19, Curve fitting methods were performed for quantitative analysis of the dynamic characteristics of nitrogen uptake and utilization process of treated rice after full heading. This sentence constitutes a broad meaning, please be specific by including specialized words that fit the topic of the research.
- P1, L 20-22, The results showed that SPAD values of the upper three leaves of rice among different treatments presented a slow-fast-slow inverted "S" curve with post-anthesis days. However, the maximum SPAD value. Why do the authors use the SPAD value for chlorophyll determination? I think that this method is inaccurate and that the device needs constant calibration and adjustment during work, and this must be clearly stated in the materials and methods section. In addition, the SPAD value does not give honest results about the content of chlorophyll in the leaves, but it is only an indicator that cannot be relied upon during the discussion of the results.
- P1, L 40-42, The results of this study provided a basis for the 40optimization of nitrogen fertilizer application at different sowing dates, real-time nitrogen fertilizer management, research, and development of post-anthesis nitrogen uptake and distribution models. So, pls be specific in your information and description of the research conclusion and recommendation at the end of the abstract section.
- The abbreviation should be used after the full term. Please be consistent with the usage of all abbreviations. Pls revise the abbreviations in the whole part of MS.
- Numerical data or ratios should be put in place to show the effects of some treatments that have given outstanding results. Pls write specific results.
- In the abstract, pls give the important results for the construction and mechanism of bacterial community for co-contamination remediation that has been estimated and studied.
Keywords: “rice; sowing date; nitrogen rate; nitrogen uptake; post-flowering dynamic models” I suggest rephrasing some words because keywords should not repeat words from the title.
- Please pay attention to the format of the manuscript.
- P2, L 49, The sowing date mainly affects the growth and development- 48ment of rice as climate changes [7,10]. I think the numbers in parentheses are [7-10] and not [7,10] according to the Journal's format, Pls pay attention to this point in the entire manuscript.
- P2, L 51-60, Ling Qihong [13] has reported that reasonable schedule of sowing date is the key technology in cultivation and management to determine the sowing date reasonably allowing to provide better light and temperature at seed setting of rice, which is the basis of high yield and good quality of rice. Nitrogen is also one of the determinants of rice growth and high yield. Studies have indicated that nitrogen uptake of rice interacts with its biomass accumulation, leaf area, and spike formation, and application of nitrogen fertilizer can increase nitrogen accumulation in above-ground dry matter, thereby increasing yield. However, excessive application of nitrogen fertilizer and low nitrogen use efficiency will reduce rice yield [14-19].
- The current state of MS needs some more attention.
- The plagiarism should be reduced according to the journal's requirements.
- Adding a short paragraph about Post-anthesis Nitrogen Dynamic Models should be added in the summary section.
- P2, L 63-65, To achieve the best nitrogen application scheme with a high yield while balancing various factors, it is necessary to continuously monitor the dynamic changes of 64post-anthesis nitrogen in rice plants. Pls add relevant ref.
- In the discussion section, conjunctions should be used to show the relationship between sentences.
- The introduction part contains long sentences, some of which are repetitive, and contains a meaning similar to what was mentioned in another paragraph made from the same part. So, pls shorten and avoid repetition.
- There is an extensive presentation of the manuscript problem at the end of the introduction part, please be direct and specific in width and thickness in the form of short and specific sentences that lead you to the goal directly
- Pls add a short paragraph about the importance and novelty of the study compared to previous studies in this regard.
- The authors did not mention anything about the design of the experiment, nor the number of transactions, nor the number of replications in the materials and methods part, neither at the beginning of this part nor at the end (in the statistical analysis part). This is a critical point.
- The materials and methods section is long; this section should be shortened as much as possible in order to reach the intended meaning directly for all the studied parameters.
- Materials and methods should be supported with references to suit the importance of this as scientific methods are approved.
- It is important to write relevant and recent refs regarding all methods used in the section of materials and methods.
- Please highlight more specifically the objective of the work.
- It is needed to re-organize and re-write the section of results and discussion. There was a lack of description for the differences among the treatments in results, but too many general statements in the discussion.
- Authors should take advantage of the chemical analysis of some components they conducted in interpreting their results, especially since these chemical components support a strong discussion of the results if they are used correctly and in the appropriate place.
- In all Tables. There are no indications of statistical analysis and comparison of averages, and from here, we cannot know the significant differences between the averages from the absence of them. Hence, the method used to compare averages and test their significance should be written in all tables.
- All data are statistically analyzed and error bar and standard deviation for provided.
- In the discussion section, conjunctions should be used to show the relationship between sentences.
- Please, make an effort to synthesize the text avoiding redundancies and repetitions in the discussion.
- The discussion section must be separated from the conclusion part, as is the case in the journal template.
- Conclusions; In conclusion, you should write a summary of your work in short sentences so that I, as a reader of this article, can understand what the article ended up being.
- Please add more discussion for more explanation especially in the important parameters as much as possible, taking into account the lack of violation of the content and the non-repetition.
- The resolution in all figures is not clear. Therefore, the reader can not know anything about the data in all figures.
- Some parts of the discussion sentences need clarification and interpretation, and recent references need to be used as much as possible.
- The discussion should be better organized and extended. It is important to try to better deepen and explain.
References;
- The number of references is about 34 ref. I think it is not enough. While, pls add the recent ones and avoid repetition. There is a recent ref. (2020-2022) in the same trend of the topic of this MS, pls pay attention to this point and cross-check all the references for mistakes, and follow the journal style of reference input.
General comments:
- The manuscript contains some typo errors; please revise it very carefully. A careful revision of the English Grammar is required. So, language needs to be improved thoroughly
- There are some references that are irrelevant to this research. Please quote the following papers published in Bite in recent years:
- There are some grammar mistakes in the manuscript.
- Hence, it is recommended to be published after major revision.

Author Response

(The authors gave the same response as above.)

Round 2
Reviewer 1 Report
The entire manuscript is written very unprofessionally. The author should take help from a technical english writer to improve the presentation of the manuscript. The language should be clear, succinct and grammatical error free to read. It doesn't generate enough interest to the reader. There are many grammatical errors and repetitive.

Author Response
Response to Reviewer Comments
Dear reviewer,
Thank you again for reviewing this manuscript in your busy schedule. Now we have carefully corrected and replied the manuscript for this revision. The language and grammar in the manuscript have been improved, and long sentences have been kept as short as possible. Words that need to be replaced are in italics. The revision instructions are as follows:
Point 1: P1, L 19: Remove the third "the" of the sentence "The results showed that the three-leaf SPAD values of rice under the different treatments varied, exhibiting a slow–fast–slow inverted "S"-shaped curve on the days after anthesis".
Response 1: The word has been removed and the sentence has been changed to " The results showed that the three-leaf SPAD values of rice under different treatments varied, exhibiting a slow–fast–slow inverted "S"-shaped curve on the days after anthesis.”
Point 2: long and unclear sentence: The leaf nitrogen content and dry matter of different treatments exhibited a linear curve, the leaf nitrogen uptake rate showed an exponential downward trend, and the decline rate and the nitrogen content, dry matter accumulation, and nitrogen uptake at the theoretical post-anthesis gradually increased with the delay of the sowing date and the increase in nitrogen; the maximum values were obtained in the S3N3 treatment
Response 2: The sentence has been changed, and as follows: Leaf nitrogen content and dry matter decreased linearly in different treatments, and leaf nitrogen uptake showed an exponential downward trend. The parameters alnc, aldm, alnu, blnc, bldm, blnu all increased gradually with the delay of sowing date and the increase of nitrogen, and the maximum values were obtained in the S3N3 treatment.
Point 3: What is grouting ?
Response 3: The grouting means a stage in which the starch, protein and accumulated organic matter produced by the crop through photosynthesis are stored in the grain through assimilation
Point 4: P2, L 46: Change “A warming climate” to “Warming climate”: A warming climate has accelerated the growth and development of rice and has shortened the growth period.
Response 4: This has been changed, please see P2, L 46.
Point 5: Is this nitrogen quality? The nitrogen status and dynamic changes in rice during flowering affect not only yield but also nitrogen quality, stress resistance, and physiological efficiency.
Response 5: I'm sorry, it should be nitrogen status.
Point 6: Remove “the” and add “In rice……”, there is no word as yield formation, and the sentences doesn’t convey meaning: The rice post-flowering period is critical for yield formation, and improving dry matter 61 accumulation and nitrogen uptake after rice flowering is the key to nitrogen use efficiency [21]. The production of post-anthesis photosynthetic products shows a relationship between the rice population status and the environment, including light and temperature. The water and fertilizer contained in the soil also undergo complex changes, and the factors described above determine the complexity and distribution process of post-anthesis nitrogen uptake.
Response 6: This has been changed, and as follows: The post-anthesis of rice is a critical period for panicle growth and development, and improving dry matter accumulation and nitrogen uptake in rice after anthesis is the key to nitrogen use efficiency [21]. The production of photosynthate after anthesis shows the relationship between population status and environment, including the complexity and distribution process of nitrogen uptake after anthesis determined by light and temperature.
Point 7: Change “predict” to “predicting” ,and this sentence is too loaded and very long. Should be broken down to simpler sentences: A crop growth model can present the basic dynamics and quantitative relationships of crop growth and the development process, quantitatively predict the dynamic behavior and yield quality of crop growth systems, assist in optimizing management and quantitative regulation, and achieve the goals of high yield, high efficiency, eco-friendly, high quality, and safety of crop production [25].
Response 7: The sentence has been changed, and as follows: The crop growth model can present the basic laws and quantitative relationships of the crop growth and development process, and quantitatively predict the dynamic behavior of the crop growth system [25].
Point 8: Its repetitive what the author is trying to say the same thing many times: At present, the analysis of rice nitrogen using a crop model is mainly based on the critical concentration of nitrogen [26,27] and nutritional diagnosis [28,29], while there are few studies on the dynamic fit of post-anthesis nitrogen under the combination of sowing date and nitrogen application. How to reasonably determine the sowing date and efficient use of nitrogen fertilizer to meet the high and stable yield of rice while achieving sustainable use of the ecological environment has become one of the problems that crop cultivators are urgently trying to solve. In this experiment, carried out in Jiangsu Province, the high-quality rice variety Nanjing 9108 was used as the material [30]. This experiment analyzed changes in the dynamic characteristics of nitrogen after anthesis of rice with different sowing dates and nitrogen application rates and their relationship with utilization efficiency.
Response 8: I'm sorry there are some repetitions in the sentence, and the sentences has been changed as follows: At present, the nitrogen model of rice is mainly based on critical nitrogen concentration [26,27] and nutritional diagnosis [28,29]. There is little research on the nitrogen dynamic model after anthesis under the combination of sowing date and nitrogen application rate. Therefore, in this experiment, carried out in Jiangsu Province, the high-quality rice variety Nanjing 9108 was used as the material [30].
Point 9: Incomplete sentence: In order to provide a basis for the study of 88 nitrogen uptake and allocation after rice anthesis using crop models.
Response 9: The sentence has been changed, and as follows: The experiment analyzes the changes of nitrogen dynamic characteristic parameters and their relationship with the utilization efficiency of rice after anthesis, and monitors and manages nitrogen absorption and utilization in real time.
Point 10: No capital letter here: China
Response 10: Sorry, China is a country name and the first letter is not capitalized.
Point 11: Be consistent with the units (g or mg): The stubble in the test field was wheat, and the soil type was sandy loam, with cultivated organic matter at 18.76 g·kg-1, total nitrogen at 1.26 g·kg-1, alkalized nitrogen at 81.76 mg·kg-1, available phosphorus at 29.26 mg·kg-1, and available potassium at 87.95 mg·kg-1 .
Response 11: Sorry there is no uniform unit symbol, and the sentence has been changed as follows: The stubble in the test field was wheat, and the soil type was sandy loam, with cultivated organic matter at 18.76 g·kg-1, total nitrogen at 1.26 g·kg-1, alkalized nitrogen at 0.08 g·kg-1, available phosphorus at 0.03 g·kg-1, and available potassium at 0.09 g·kg-1.
Point 12: Is that Ca pyrophosphate ? and is this area? Or size? The sentence is “Prior to sowing, calcium persophosphate 450 kg·hm-2 and potassium chloride 150 kg·hm-2 were applied to each treatment.”
Response 12: There are two problems here. The answer to the first question is that here is calcium superphosphate. The answer to the second question is that the data here express the application of phosphorus and potassium fertilizers for each treatment. The sentence has been changed, and as follows: Prior to sowing, calcium superphosphate 450 kg·hm-2 and potassium chloride 150 kg·hm-2 were applied to each treatment.
Point 13: Very carelessly written sentences: The nitrogen fertilizer for the test was urea, and ratio of the basal fertilizer: first tillering fertilizer: second tillering fertilizer: flower-promoting fertilizer: and flower-preserving fertilizer was 3:1.5:1.5:2.5:1.5.
Response 13: The sentence has been changed, and as follows: The nitrogen fertilizer tested was urea, and the fertilizer ratio of basis: first tiller: second tiller: flower promotion: flower preservation was 6:3:3:5:3.
Point 14: With or between, can not use both. What is this hole? We transplanted seedlings with between 3.4 and 3.8 leaves, with a plant spacing of 30 cm×13 cm and four seedlings per hole.
Response 14: There are two problems here. The answer to the first question is to delete between. The answer to the second question is that the hole means the hill. The sentence has been changed as follows: We transplanted seedlings with 3.4 and 3.8 leaves, with a plant spacing of 30 cm×13 cm and four seedlings per hole.
Point 15: Doesn't convey the meaning: Field management of the whole experimental process was the same as that used in field production.
Response 15: Sorry, and the sentence has been changed as follows: Other field management in the test process was the same as field production.
Point 16: Unclear: Each treatment was calibrated with 10 representative main stems; every seven days from the beginning of the rice panicle stage, the SPAD values of three leaves were measured with the SPAD-502 chlorophyll meter produced by Konica Minolta Sensing, Inc., and the middle of each leaf (avoiding the main leaf vein) was measured as the measurement point until maturity [31].
Response 16: Sorry, this sentence is not clear. The sentence has been changed as follows: A representative 10 main stems of rice were measured for each treatment. Every 7 days after anthesis, measure the SPAD value of three leaves with a SPAD-502 chlorophyll meter manufactured by Konica Minolta, measuring the middle of each leaf (avoiding the main veins) as until maturity[31].
Point 17: Please make short and clear sentences in the whole document: After the rice panicle stage, we collected the plants from the representative four holes in each cell every seven days. We divided each plant into three parts: stem-sheath, leaf, and grain, placed each sample in an oven at 105 °C for 30 minutes, and then dried the plant tissue at 80 °C to a constant weight. After cooling to room temperature in a dry environment, the dry matter of the ear sheath, leaf, and grain was measured separately, and the dry matter of each organ per unit area was calculated according to the average number of panicles obtained from the 60-hole survey in each cell.
Response 17: This document has been shortened and clear. The sentence has been changed as follows: After rice anthesis, four representative holes were taken every seven days for each treatment, and the plants were divided into three parts: stem-sheath, leaf and grain. The samples were placed in an oven at 105°C for 30 minutes, and then dried at 80°C to constant weight. After cooling to room temperature in a dry environment, the dry matter was measured.
Point 18: Change in SPAD values of inverted sword leaves (days after anthesis). Change in all the figures
Response 18: Changed, and all the figures have been changed.
Point 19: Too long sentence: As shown in Table 1, the coefficient of determination of each treatment curve exceeded 0.987, indicating that the curve fitting results were ideal, and that this model accurately described the change process of the upper three leaves SPAD value of rice with the growth time after anthesis.
Response 19: This sentence has been shortened, and as follows: There were significant differences in the main parameters of dynamic changes in the SPAD value of the upper three leaves after anthesis, as shown in Table 1. The coefficient of determination R2 of each treatment curve was above 0.987, indicating that the curve fitting results were ideal.
Point 20: Change “The change in the curve of the stem….. “ to “The change curve of the stem-sheath nitrogen content with days after anthesis is shown in Figure 4.”
Response 20: The sentence has been changed as follows: The change in the curve of the stem-sheath nitrogen content with days after anthesis is shown in Figure 4.
Point 21: Change “along with the ........... “ to “The grain nitrogen uptake rate changed over time as did the grain dry matter, showing a "slow–fast–slow" S-shaped upward trend.”
Response 21: The sentence has been changed as follows: The grain nitrogen uptake rate changed over time along with the grain dry matter, showing a "slow–fast–slow" S-shaped upward trend.
Point 22: Make 2-3 paragraphs of discussion.
Response 22: The discussion has been divided into three paragraphs, and as follows:
Dry matter accumulation is the material basis for yield formation during crop growth, and the dry matter accumulation of rice under different treatments differs; quantitative analysis of the dynamic changes in dry matter accumulation during crop growth is of great significance to reveal crop yield formation [33]. Nitrogen uptake in rice varies with growth, and models can quickly and easily estimate nitrogen needs for rice growth [28]. In this experiment, the dry matter accumulation and nitrogen accumulation processes of each organ of different rice plants were fitted by different models, the relevant characteristic parameters in the fitting equation were calculated. In addition, the dynamic characteristics of dry matter accumulation and nitrogen uptake and allocation processes were quantitatively analyzed.
The SPAD value was significantly correlated with the leaf nitrogen content, which was a sensitive indicator reflecting the dynamic change in nitrogen of plants, and the nitrogen status during rice growth was monitored, and the nitrogen application was guided by nitrogen application to obtain high yields and improve nitrogen use efficiency [34-36].The results of this study show that the change trend in the SPAD values of the upper three leaves after anthesis was an inverted S-shaped downward curve, which was basically consistent with the study of Zhao et al. [37]. In contrast, some studies found that the SPAD values of the upper four leaves after anthesis of rice linearly decreased [38], which is different from the results of this study and may be due to significant differences in the nitrogen application rate or the varieties or settings selected for the experiment. The results of this study show that the duration of the maximum SPAD value (ks) and rapid descent period (t2s-t1s) in each of the upper three leaves increased with the increase of nitrogen fertilizer, indicating that the more nitrogen applied, the longer and slower the rapid decline of nitrogen in leaves. This is the same as the result of the gradual increase of the SPAD value of leaf with the increase of nitrogen application in Huang et al. [39]. In this study, the leaf nitrogen content and dry matter showed a linear downward trend, and the nitrogen uptake rate showed an exponential downward trend, similar to the results reported by Xu et al. [40]. Cao et al. [41] conducted four studies that showed that the change trend in nitrogen uptake of post-anthesis grains at different sowing periods and nitrogen application rates could be dynamically fitted using the NRMSE model, and the fitting effect was good. The change trend in the dry matter and nitrogen uptake in the grain in this study was an S-type upward curve, and the fitting effect of the inverted logistic model was excellent. That is inconsistent with previous research results and may be due to certain differences in external factors such as the experimental environment and cultivation measures. The change trend in dry matter and nitrogen uptake was a parabolic line with an upward-pointing opening, which is basically consistent with the results of Xu [42].
The dry matter and nitrogen uptake of the stem-sheath and leaf of the late sowing were higher than those of other treatments, and the dry matter weight and nitrogen uptake of the grain entered the rapid decline rate (Tlnu) at the earliest. This was inconsistent with Sun et al. [43] study that the contribution rate of the stem-sheath and leaf dry matter to the panicle, and the contribution rate of the stem-sheath and leaf nitrogen to the panicle were all higher than the suitable sowing date. Pal et al. [44] found that the source activity of late-sown rice during the reproductive growth was limited, resulting in low dry matter accumulation and nitrogen uptake during flowering, and increased dry matter accumulation after anthesis, coupled with increased nitrogen uptake and transport during flowering, led to an increase in reservoir capacity. However, in this experiment, the dry matter accumulation and nitrogen uptake of rice after late sowing were higher than those of other treatments, and they increased with the increase of nitrogen fertilizer. The research results were inconsistent with previous studies, which was likely to be affected by the interaction between rice varieties, regional environment, sowing date and nitrogen application rate. The S1 and S2 treatments had higher nitrogen uptake at the N2 level and the S3 treatment had higher nitrogen uptake at the N3 level. This was consistent with the results of Stone et al. [45] study that late sowing could reduce nitrogen loss, indicating that late sowing could increase the source capacity and reservoir capacity at high nitrogen levels, and promote the absorption and accumulation of nitrogen by plants. This study fills the gap in this regard by making up for the lack of attention to nitrogen dynamics in the aboveground plant parts except after wheat anthesis [46,47]. There are few reports on the nitrogen dynamics after flowering in rice; hence, this study lays a foundation for the study of nitrogen dynamics after anthesis.
Point 23: You already mentioned this in the methodology, and rewrite the sentence: The only rice variety studied was Nanjing 9108, and whether the results obtained are applicable to other types of varieties needs further study.
Response 23: This sentence has been deleted.
Thank you for your consideration. We greatly appreciate your comments, as your comments make our research perfect. I look forward to hearing from you.
Sincerely,
Ying Ye
April.9,2022

Reviewer 2 Report
The introduced corrections are satisfactory for me.
Author Response
Dear reviewer,
Thank you for your affirmation and for taking the time to review this manuscript. We greatly appreciate your comments, as your comments make our research perfect.
We hope to receive your approval. We look forward to hearing from you.
Sincerely,
Ying Ye
April.9,2022

Reviewer 4 Report
Sustainability-MDPI
MS No.: sustainability-1634069
Manuscript Title: "Post-anthesis nitrogen dynamic models and characteristics of rice combined with sowing date and nitrogen application rate"
Reviewer comments (R2):
The comments:
After reviewing the entire manuscript and revising my comments on the manuscript, I found that the manuscript has improved now, but the following must be taken into account:
- The quality of the graphs is still not good and needs further clarification to increase its quality.
- The references should be reviewed carefully, especially after the modifications made in the entire manuscript
- The manuscript contains some typo errors. A careful revision of the English grammar is required. In addition, the manuscript needs more linguistic revision to increase the quality of its scientific writing, hence, it is recommended to be published after minor revision.

Author Response
Dear reviewer,
Thank you very much for your affirmation and for taking the time to read my manuscript. Now we have carefully corrected and replied the manuscript for this revision. The revision instructions are as follows:
Point 1: The quality of the graphs is still not good and needs further clarification to increase its quality.
Response 1: I'm sorry that the picture you see is not clear enough, but the picture is already the highest definition. If you cannot see clearly, please enlarge the picture without reducing the clarity of the picture.
Point 2: The references should be reviewed carefully, especially after the modifications made in the entire manuscript
Response 2: Thank you very much for the reminder, and the references have been carefully checked.
Point 3: The manuscript contains some typo errors. A careful revision of the English grammar is required. In addition, the manuscript needs more linguistic revision to increase the quality of its scientific writing, hence, it is recommended to be published after minor revision.
Response 3: The manuscript has been re-edited in English by professionals in the industry on the MDPI official website. Please review the revised manuscript.
Thank you for your consideration. We greatly appreciate your comments, as your comments make our research perfect. I look forward to hearing from you.
Sincerely,
Ying Ye
April.9,2022

Round 3
Reviewer 1 Report
There are a few minor edits that needs to be take care of. The authors have revised the document and its in a good presentable form now.

Author Response
Dear reviewer,
Thank you again for taking the time to review this manuscript so carefully. We have now carefully corrected and responded to this revised manuscript. Words in italics are modified parts.The revision notes are as follows:
Point 1: P2, L 82: avoid capitalizing the third "the" of the sentence " The test was conducted from May to November 2018 at the Experimental Farm of The College of Agronomy of Yangzhou University, Yangzhou city, Jiangsu Province, China (119°25'E, 32°23'N), with an average annual precipitation of 1288 mm and an average sunshine duration of 1973.9 hours. ".
Response 1: The word has been revised and the sentence has been changed to " The test was conducted from May to November 2018 at the Experimental Farm of the College of Agronomy of Yangzhou University, Yangzhou city, Jiangsu Province, China (119°25'E, 32°23'N), with an average annual precipitation of 1288 mm and an average sunshine duration of 1973.9 hours.”
Point 2: spacing needs to be removed.
Response 2: Thank you for your reminder. The entire article has been formatted with spacing changed as required by the journal.
Thank you for your consideration. We greatly appreciate your comments, as your comments make our research perfect. I look forward to hearing from you.
Sincerely,
Ying Ye
April.14,2022
